# Obesity drives depot-specific vascular remodeling in male white adipose tissue

Sana S. Hasan[1], David John [ID]2,3, Martina Rudnicki [ID]4, Ibrahim AlZaim [ID]5, Daniel Eberhard[6], Iris Moll[7], Jacqueline Taylor[7], Christian Klein [ID]8, Maximilian von Heesen[9], Lena-Christin Conradi[9], Ralf H. Adams [ID]10, Eckhard Lammert [ID]6,11,12, Joanna Kalucka [ID]5, Christiana Ruhrberg [ID]4, Stefanie Dimmeler [ID]2,3,13 & Andreas Fischer [ID]1,14 ✉

Obesity-driven pathological expansion of white adipose tissue (WAT) is a key driver of endothelial dysfunction. However, early vascular alterations associated with over-nutrition also serve to exacerbate WAT dysfunction. Here, we conduct a single-cell transcriptomic analysis of WAT endothelium to delineate endothelial heterogeneity and elucidate vascular alterations and its consequence in a male murine model of obesity. We demarcate depot-specific differences in subcutaneous (sWAT) and visceral WAT (vWAT) endothelium through in sillico analysis and further corroboration of our findings. Moreover, we identify a sWAT-specific fenestrated endothelial cell (EC) subtype, which declines in obese conditions. Utilizing systemic anti-VEGFA blockade and genetic *Vegfa* manipulation, we demonstrate that VEGFA is necessary for maintaining fenestration in sWAT. Additionally, we detect this fenestrated EC subtype in male human WAT, which undergoes reduction in individuals with obesity. Collectively, this atlas serves as a valuable tool for future studies to decipher the functional significance of different WAT EC subtypes.

Obesity is a major health concern, as it is associated with several comorbidities including dyslipidemia, hypertension, cardiovascular disease, insulin resistance, type-2 diabetes, steatohepatitis, and certain types of cancer[1]. White adipose tissue (WAT) plays a pivotal role in regulating systemic energy metabolism. It serves as a safe storage site for excess lipids and secretes several adipokines[2], which signal the functional status of WAT to other organs. As obesity progresses, rampant expansion of adipocytes in WAT leads to maladaptive tissue remodeling, resulting in chronic inflammation and an imbalanced adipokine secretome[3]. Although excessive adiposity is generally detrimental to health, the metabolic outcome is strongly associated with the specific anatomical location of fat deposition. WAT is distributed in two large anatomical locations: subcutaneous WAT (sWAT) and visceral WAT (vWAT). Accumulation of fat in vWAT correlates with poor metabolic health, whereas deposition in sWAT poses lower risk for metabolic diseases. These observations highlight

[1]Department of Clinical Chemistry, University Medical Center Göttingen, Göttingen, Germany. [2]Institute of Cardiovascular Regeneration, Goethe University, Frankfurt, Germany. [3]Cardio-Pulmonary Institute (CPI), Frankfurt, Germany. [4]UCL Institute of Ophthalmology, University College London, London, UK. [5]Department of Biomedicine, Aarhus University, Aarhus, Denmark. [6]Heinrich Heine University Düsseldorf, Faculty of Mathematics and Natural Sciences, Institute of Metabolic Physiology, Düsseldorf, Germany. [7]Division Vascular Signaling and Cancer, German Cancer Research Center (DKFZ), Heidelberg, Germany. [8]Roche Pharma Research and Early Development, Discovery Oncology, Roche Innovation Center Zurich, Roche Glycart AG, Schlieren, Switzerland. [9]Department of General, Visceral and Pediatric Surgery, University Medical Center Göttingen, Göttingen, Germany. [10]Department of Tissue Morphogenesis, Max Planck Institute for Molecular Biomedicine, Münster, Germany. [11]German Diabetes Center, Leibniz Center for Diabetes Research at Heinrich Heine University Düsseldorf, Düsseldorf, Germany. [12]German Center for Diabetes Research (DZD e.V.), Neuherberg, Germany. [13]German Center for Cardiovascular Research (DZHK), Berlin, Germany. [14]German Center for Cardiovascular Research (DZHK), partner site Lower Saxony, Saxony, Germany. ✉e-mail: andreas.fischer@med.uni-goettingen.de

the underlying biological and functional differences between the two WAT depots[4]. Despite the fact that adipocytes constitute the majority of the adipose tissue, recent scientific discoveries have revealed that the tissue's health and functionality are tightly regulated through intricate crosstalk between the various cell types present within the tissue[5].

Adipose tissue is highly vascularized, with each adipocyte in close contact with a capillary[6]. Endothelial cells (ECs), which form the inner lining of all blood vessels, serve not only as conduits for transport but have also been identified as crucial mediators of organ function, facilitating intercellular communication through angiocrine factors[7,8]. During the course of obesity, rapidly expanding hypoxic adipose tissue in combination with inflammation and fibrosis drives endothelial dysfunction and vascular rarefaction. This suggests that vascular impairment is a consequence of obesity[3]. However, contrary to the prevailing view, there is also evidence that early vascular alterations associated with overnutrition may exacerbate adipose tissue dysfunction[9]. It is therefore crucial to dissect the precise relationship between endothelial dysfunction and obesity.

Recent advances in single-cell transcriptomics have led to the discovery of novel EC types and heterogeneity in different organs. A recent study has described organ-specific endothelial vulnerabilities during in obesity[10]. Nevertheless, a more detailed insight into the precise mechanism of how obesity alters endothelial heterogeneity remains elusive.

In this work, we utilize single-cell RNA sequencing (scRNA-seq) in conjunction with Cell Hashing methodology to generate a comprehensive map of murine adipose tissue-derived endothelial transcriptome at a single-cell resolution under lean and obese conditions. We delineate key depot-specific differences in sWAT and vWAT endothelium. Furthermore, we identify a sWAT-specific fenestrated endothelial cell (EC) subtype, which declines in obese conditions. Employing systemic anti-VEGFA blockade and genetic *Vegfa* manipulation, we demonstrate that VEGFA is necessary for maintaining fenestration in sWAT. Additionally, we detect this fenestrated EC subtype in male human WAT, which undergoes reduction in individuals with obesity. Together, this atlas serves as a valuable tool for future studies to decipher the functional significance of different WAT EC subtypes.

## Results

### Vascular rarefaction in obese WAT

Given the differences in the ontogeny and function of different WAT depots, we conducted an ultrastructural analysis of sWAT and vWAT and found more collagen fibers in sWAT versus vWAT (Supplementary Fig. 1a), indicative of structural differences between these WAT depots. To gain insight into the progression of vascular remodeling in different WAT depots under obese conditions, we performed immunofluorescent staining for pan-endothelial marker CD31 on gross tissue sections from diet-induced murine model of obesity (Fig. 1a). Mice were fed either a control (CD) or a high-fat diet (HFD) for eight weeks leading to a significant difference in body weights between the two groups at the end of the feeding period (Supplementary Fig. 1b).

To assess whether the vascular network is able to cope up with the expanding tissue, we evaluated vascular density and observed a significant reduction in vascular density in both sWAT and visceral vWAT depots (Fig. 1b, c). Notably, even under lean conditions blood vessel density was markedly higher in sWAT than in vWAT (Fig. 1d). Moreover, in obese mice, we observed an increase in p53+ EC nuclei in large caliber vessels (arrowheads, Supplementary Fig. 1c-f), previously shown to be associated with endothelial dysfunction under various conditions[11,12]. We also observed a depot-specific decrease in perivascular coverage in vWAT (Supplementary Fig. 1g-j).

### The single-cell transcriptome of adipose tissue-derived endothelial cells

To gain a better understanding of how obesity alters endothelial heterogeneity in WAT, we conducted a high-throughput droplet-based single-cell transcriptomics analysis on viable CD31 + CD45- ECs from vWAT and sWAT under lean and obese conditions (Supplementary Fig. 2 a-c). We also incorporated Cell Hashing[13] method to label cells isolated from each biological replicate with a uniquely barcoded Hashtag antibody (Supplementary Fig. 2d, Supplementary Table 1). This approach enabled the data to be demultiplexed at a later stage, facilitating the application of more sophisticated statistical techniques, as each transcript could be assigned to the mouse from which it was obtained.

Following rigorous quality control procedures, including the assignment of appropriate hashtags to cells and the removal of doublets, we recovered approximately 14,000 cells (Supplementary Table 2, 3) and identified 12 distinct clusters (Fig. 1e). Each of these clusters were represented in all three biological replicates (Supplementary Fig. 2 e-h). We identified seven EC-specific clusters based on the expression of well-established pan-endothelial markers including *Pecam1* (encoding CD31) and *Cdh5* (Supplementary Fig. 3a, Supplementary Table 4-10). Additionally, some non-EC types were identified based on the expression of fibroblast (*Dcn* and *Lmn*) and mural cell markers (*Pdgfrb* and *Acta2*) (Supplementary Fig. 3a). No lymphatic (*Lyve1* and *Prox1*) or hematopoietic clusters (*Ptprc* and *Hbb-bs*) were identified in the dataset (Supplementary Fig. 3a). Cell types were classified based on the expression of cluster-specific marker genes (Fig. 1f, Supplementary Fig. 3b) and previously published literature[14]. In this study, our focus has been on the clusters depicted in Fig. 1e.

We identified two distinct capillary types (Cap I and II), enriched in markers involved in uptake and metabolism of fatty acids and glycerol (*Gpihbp1, Fabp4, Tcf15, Cd36*) (Fig. 1f, Supplementary Fig. 4a, b), concordant with the organ-specific function of adipose tissue vasculature[6]. Furthermore, we identified another capillary population (Cap III), expressing high levels of extracellular matrix genes (Fig. 1f, Supplementary Fig. 4c).

Venous ECs were annotated based on the presence of canonical markers like *Ackr1* and *Selp* (Fig. 1f, Supplementary Fig. 4d). ECs in the arterial cluster were identified based on the expression of known arterial markers, including *Fbln5*, *Gja4* and *Stmn2*, as well as Notch downstream target *Hey1* (Fig. 1f, Supplementary Fig. 4e). Moreover, arterial ECs expressed higher levels of *Sema3g*, a recently described arterial marker[14], along with other notable markers such as *Vegfc* and *Col8a1* (Supplementary Fig. 3b). To validate our transcriptome analysis, we performed whole-mount immunofluorescent staining for arterial (GJA5, arrowheads, Supplementary Fig. 4f) and venous markers (P-Selectin, arrow, Supplementary Fig. 4f), which showed non-overlapping expression.

Moreover, we identified a cluster with an angiogenic phenotype, exhibiting tip-cell marker genes like *Apln*[15] (Supplementary Fig. 5a). To gain insight into the potential biological role of this cell type, we performed pathway analysis on top marker genes within this cluster. Additionally, hallmarks for epithelial-to-mesenchymal transition and apical junction were enriched along with PI3K/AKT signaling (Supplementary Fig. 5b), which are crucial processes regulating endothelial barrier function, polarity and migration[16,17]. Similarly, Ingenuity Pathway Analysis (IPA) revealed an invasive and migratory phenotype (red asterisk, Supplementary Fig. 5c), indicating that these ECs exhibit a highly angiogenic characteristic.

Lastly, we discovered an EC subtype in WAT that exhibited high expression for known markers of fenestrations[18] (*Plvap, Esm1*) (Fig. 1f, Supplementary Fig. 5d). Additionally, *Itm2a* (previously described to be mostly restricted to brain capillaries[19]) was also enriched in this cluster (Supplementary Fig. 5d). We could validate these findings by whole-mount immunostaining showing non-overlapping mutually

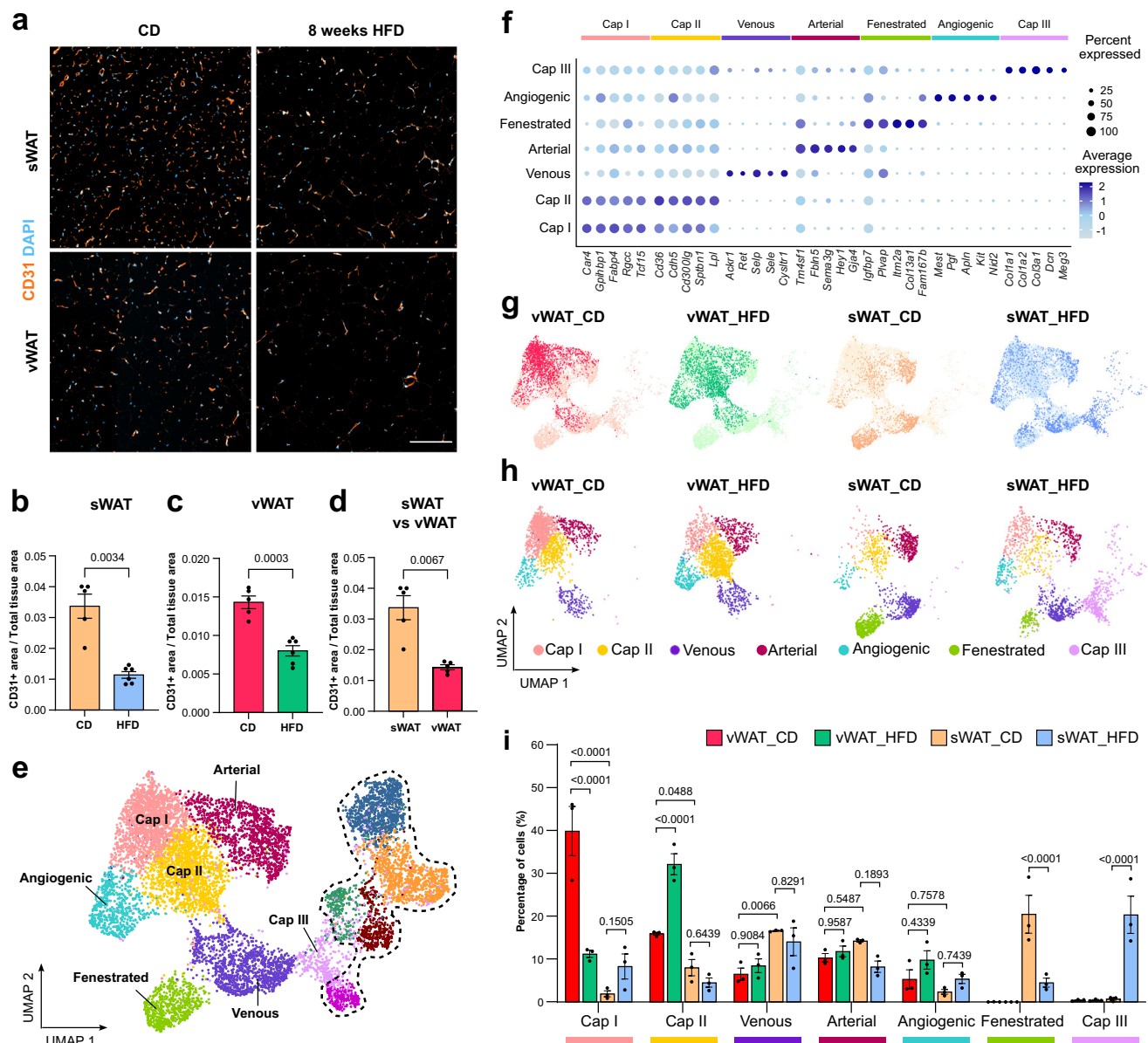

**Fig. 1 | Single cell transcriptome of adipose tissue-derived endothelial cells.**
**a** Representative confocal images showing CD31+ (orange) blood vessels and DAPI+ (cyan) nuclei in subcutaneous white adipose tissue (sWAT) or visceral WAT (vWAT) sections from mice fed either a control diet (CD) or high fat diet (HFD) for 8 weeks. **b** Quantification of blood vessel (CD31+) area in sWAT (CD $n = 5$, HFD $n = 6$ mice). **c** Quantification of blood vessel (CD31+) area in vWAT (CD $n = 5$, HFD $n = 6$ mice). **d** Comparison of vessel density in lean sWAT vs vWAT (sWAT $n = 5$, vWAT $n = 5$ mice). **e** Uniform manifold approximation and projection (UMAP) clustering of cells obtained from sWAT and vWAT from CD and HFD fed mice (dotted line represents clusters not discussed further). **f** Dot plot depicting top enriched marker genes in annotated clusters. The color intensity of each dot represents the level of marker expression, whereas the dot size reflects the percentage of endothelial cells (ECs) expressing the marker within the EC subcluster. **g** Demultiplexed UMAP of ECs in different depots in lean (CD) and obese mice (HFD). **h** Differential clustering of ECs in different depots and conditions. **i** Percentage of cells belonging to each annotated cluster under different condition and depots (CD $n = 3$, HFD $n = 3$ mice). Scale bar 200 µm. Data represents ± SEM, two-sided Welch's t-test (**b**–**d**); two-way ANOVA with Tukey's multiple comparison test (**i**). Source data are provided as a Source Data file.

exclusive expression of PLVAP (arrowheads, Supplementary Fig. 5e) and capillary marker CAR4 (arrows, Supplementary Fig. 5e).

## Depot-specific changes in endothelial heterogeneity during obesity

In order to gain insight into the impact of obesity driven alterations in different WAT depot endothelium, we demultiplexed the data and observed cluster-specific variations in the transcriptional landscape of WAT ECs, both in depot-specific as well as in diet-specific manner (Fig. 1g, h). For instance, Cap I and Cap II ECs were present at a higher percentage in vWAT than in sWAT (Fig. 1i). Furthermore, Cap I ECs

were more prevalent in lean mice compared to obese mice, which had a higher percentage of Cap II ECs (Fig. 1i).

ECs in larger caliber vessels also showed some distinctive patterns. The percentage of venous ECs was higher in lean sWAT when compared to lean vWAT (Fig. 1i). Furthermore, we observed a trend towards decrease in percentage of arterial ECs in obese sWAT when compared to lean sWAT (Fig. 1i). Lastly, we identified EC populations that are exclusive to sWAT. For instance, fenestrated ECs were exclusively present in sWAT and were strongly reduced in obese mice (Fig. 1i). Whereas, Cap III ECs were limited to sWAT and predominantly observed in obese tissue (Fig. 1i).

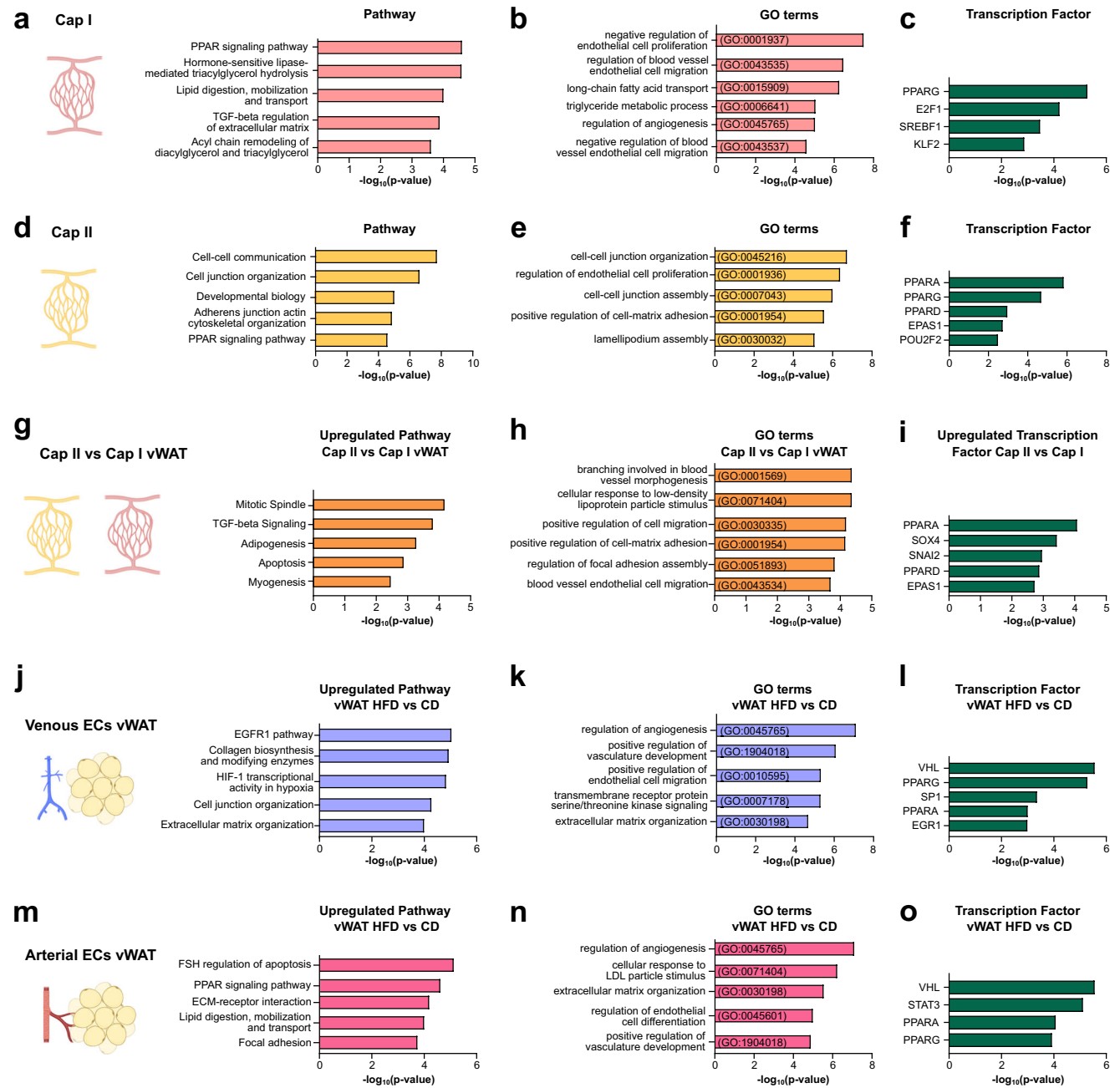

**Fig. 2 | Obesity associated vascular morphogenesis and remodeling in vWAT.**
**a** Bioplanet annotated pathways upregulated in Cap I endothelial cells (ECs). **b** Gene ontology (GO) analysis showing enriched biological processes in Cap I ECs. **c** TTRUST transcription factor (TF) analysis predicting upstream regulators in Cap I ECs. **d** Bioplanet annotated pathways upregulated in Cap II ECs. **e** GO term analysis showing enriched biological processes in Cap II ECs. **f** TTRUST TF analysis predicting upstream regulators in Cap II ECs. **g** Bioplanet predicted pathways upregulated in Cap II when directly compared to Cap I in visceral white adipose tissue

(vWAT) ECs. **h** GO terms enriched in Cap II ECs when compared to Cap I vWAT ECs. **i** TTRUST TF analysis depicting differentially enriched regulators in Cap II vWAT ECs when compared to Cap I vWAT ECs. **j** Upregulated pathways **k** enriched GO terms and **l** predicted TFs in venous ECs in obese (high fat diet, HFD) vs lean (control diet, CD) vWAT. **m** Upregulated pathways **n** enriched GO terms and **o** predicted TFs in arterial ECs in obese (HFD) vs lean (CD) vWAT. Fisher's exact test (a-o). All schematics created in BioRender. Hasan, S. (2025) https://BioRender.com/ossmape. Source data are provided as a Source Data file.

## Obesity-associated vascular morphogenesis and remodeling in vWAT

We first focused on obesity-driven changes in capillary beds in vWAT. Pathway analysis of Cap I-specific top marker genes revealed an enrichment in processes associated with lipid handling and fatty acid metabolism (Fig. 2a). Notably, gene ontology (GO) analysis indicated an active negative regulation of EC proliferation and migration. This finding highlights that endothelial quiescence is an actively regulated process[20] (Fig. 2b). Furthermore, transcription factor (TF) analysis

revealed activity of TFs with established roles in regulating lipid and glucose metabolism (*Pparg, Srebf1*)[21,22] and maintaining quiescence (*Klf2*)[23], which correlates with the pathway analysis (Fig. 2c).

Next, pathway analysis of Cap II-specific top marker genes revealed an enrichment in processes associated with cell-cell communication and junctional reorganization (Fig. 2d, e). We observed a shift in the TF profile with *Ppara* being the most upregulated TF (Fig. 2f). *Ppara* is mainly expressed in liver and promotes fatty acid uptake[24]. However, the endothelial *Ppara* upregulation may be

indicative of alterations in the lipid profiles associated with obesity. Based on differentially expressed genes (DEGs), a comparison of the two capillary subtypes in vWAT revealed enrichment of pathways associated with cholesterol metabolism and junctional assembly in Cap II (Fig. 2g) and downregulation of processes involved in cellular respiration in Cap II (Supplementary Fig. 6a, b). The most significant processes that were upregulated in Cap II vWAT ECs were related to blood vessel morphogenesis, EC proliferation and migration (Fig. 2h), indicating that tissue expansion associated with obesity activates a vascular augmentation program in vWAT ECs. Additionally, we observed a predicted increase in *Sox4* activity (Fig. 2i), which has been described as being upregulated in response to hyperlipidemia and hyperglycemia, promoting endothelial to mesenchymal transition, inflammation, and angiogenesis in pathological settings[25,26].

To comprehend the impact of obesity on different large-caliber vessels, we conducted a subsequent investigation into the expression of upregulated DEGs in venous and arterial ECs within vWAT of obese mice. Analysis of upregulated DEGs in venous ECs in obese vWAT when compared to venous ECs in lean vWAT exhibited enrichment of pathways related to EGFR signaling, collagen synthesis and extracellular matrix organization along with hypoxia-induced *Hif1a* transcriptional activity (Fig. 2j). Although *Hif1a* activity is principally associated with induction of VEGF signaling[27], previous studies have demonstrated that HIF-1α also induces fibrosis in vWAT, which is characterized by excessive deposition of extracellular matrix (ECM) and vascular remodeling[28]. Additionally, we observed a concomitant enrichment of GO terms associated with vascular development and remodeling (Fig. 2k) and predicted *Vhl* activity (Fig. 2l), which is a master regulator of HIF-1α activity.

The trend observed in arterial vWAT ECs was analogous to that seen in venous vWAT ECs, with pathways and GO terms for ECM organization and vascular development being upregulated in obese vWAT arterial ECs (Fig. 2m, n), while processes related cytoplasmic translation were downregulated (Supplementary Fig. 6c, d). Furthermore, the induction of lipid trafficking pathways and processes was observed during obesity (Fig. 2m, n). In addition to *Vhl*, *Stat3* was also activated in obese vWAT arterial ECs (Fig. 2o). STAT3 signaling is induced in response to oxidative stress and exacerbates endothelial dysfunction[29]. Taken together, obesity induces remodeling of both micro and macrovasculature in vWAT through vascular morphogenesis and ECM reorganization.

## Obesity-driven vascular alterations in sWAT

Next, we wanted to discern obesity-induced depot-specific vascular alterations in sWAT. We identified a capillary subtype Cap III, which was exclusively present in obese sWAT. Cap III ECs were characterized by elevated transcript levels of *Col1a1*, *Col1a2* and *Col3a1* (Fig. 3a), a finding that was validated by qPCR analysis of freshly isolated sWAT ECs from lean and obese mice (Fig. 3b). Based on top marker gene analysis, the principal pathways and processes that are upregulated in this cluster have been linked to ECM biosynthesis and organization, as well as integrin interactions (Fig. 3c, d). It was reported that ECs transiently upregulate ECM component genes and acquire a mesenchymal phenotype to facilitate vascular repair. However, sustained activation of these processes can lead to fibrosis and endothelial dysfunction[30]. Furthermore, we identified *Stat6* and *Myc* as probable upstream regulators (Fig. 3e).

Next, to analyze differences in large-caliber vessels, we identified upregulated DEGs in venous ECs in obese sWAT when compared to venous ECs in lean sWAT. Obese sWAT venous ECs exhibited an interferon response, accompanied by an enrichment in cholesterol and fatty acid metabolism (Fig. 3f). IFN-α and IFN-γ are key mediators of inflammatory responses and endothelial dysfunction in obesity[31,32]. In addition, venous ECs in obese sWAT display activation of RAGE pathway (Fig. 3f). It has been demonstrated that advanced glycation

end products (AGEs) accumulate in a number of physiological processes, including aging, oxidative stress, hyperglycemia, and inflammation. RAGE activation is a hallmark of stressed and damaged cells and drives pathogenesis in numerous disease settings including vascular injury[33,34]. Furthermore, obese sWAT venous ECs were enriched in markers for platelet aggregation (Fig. 3g) that have been linked to endothelial dysfunction in metabolic disorders[35]. In addition, obese sWAT venous ECs demonstrated enhanced *Foxo1* expression (Fig. 3h), which has been shown to be associated with impaired insulin sensitivity and eNOS activation in patients with obesity[36]. *Stat1* expression was also predicted to be upregulated in obese sWAT venous ECs, which is a key mediator of IFN-γ signaling[37] (Fig. 3h).

Similar DEG based analysis in sWAT arterial ECs showed an enrichment for mesenchymal transition, along with a pathway involved in signaling of the inflammatory cytokine oncostatin M[38] in obese sWAT arterial ECs (Fig. 3i). Sema4D signaling was upregulated in both venous and arterial ECs in obese conditions, and is known to drive endothelial dysfunction in pathological settings[39], while cytoplasmic translation was downregulated in both venous and arterial subtypes in obese sWAT (Supplementary Fig. 6e, f). Apart from several changes in cellular components (Fig. 3j), there was predicted *Parp1* activity (Fig. 3k), known to be induced by DNA damage, associated with oxidative and nitrosative stress[40]. Intriguingly, *Erg* activity was also predicted to be increased in arterial ECs in obesity (Fig. 3k), which has been reported to be a crucial regulator of vascular inflammation[41].

## Presence of fenestrated endothelium exclusively in sWAT
Based on their barrier properties, the endothelium can be continuous (e.g., heart, brain), fenestrated (e.g., kidney, endocrine organs), or sinusoidal (e.g., liver, bone marrow)[7] to orchestrate trans-endothelial transport processes[42]. The endothelium of WAT is typically described as continuous, with only a couple of studies indicating the presence of fenestrated endothelium[10,43]. Our WAT transcriptome analysis identified an EC cluster that expressed prominent markers of fenestration with high congruency with choroid plexus ECs with similar enrichment of *Plvap*, *Esm1* and *Nrp1*[14] (Fig. 4a). This EC subtype was identified exclusively in sWAT and is strongly reduced in obese tissue. Bulk mRNA analysis of freshly isolated ECs from lean and obese sWAT revealed a comparable reduction in marker genes identified in this cluster (Fig. 4b). A similar analysis of vWAT ECs did not show any differences (Supplementary Fig. 7a). As *Esm1* expression was restricted to fenestrated ECs, we used *Esm1* reporter mice to label this cell population in lean mice. *Esm1* + ECs (cyan, arrowheads, Fig. 4c) showed a perfect co-localization with PLVAP (magenta, Fig. 4c). Furthermore, at an ultrastructural level, ECs freshly isolated from WAT develop fenestrae (~50 nm in diameter) that were clustered in groups after ex vivo cultivation for 12 h (Fig. 4d). While some *Esm1*+ cells were observed in vWAT, they did not show any PLVAP expression (arrowheads, Supplementary Fig. 7b).

## Correlation between VEGF signaling and presence of fenestrated endothelium
To elucidate the signaling process involved in this vascular bed, we performed pathway and GO term analysis on cell type-specific top marker genes. Neuropilin interactions with VEGF signaling components being the top enriched terms (Fig. 4e, f). Transcription factor analysis predicted *Epas1* as an upstream regulator (Fig. 4g). EPAS1 has been shown to activate both VEGFA and VEGFR2 in ECs, albeit in the context of angiogenesis[44]. To further substantiate our results we performed IPA analysis. Some of the principal nodes identified within the signaling network were associated with VEGFA, cell survival and viability (asterisk, Fig. 4h).

Next, we examined the levels of *Vegf* receptors (*Vegfr*) in our dataset and found that fenestrated ECs exhibited elevated levels of

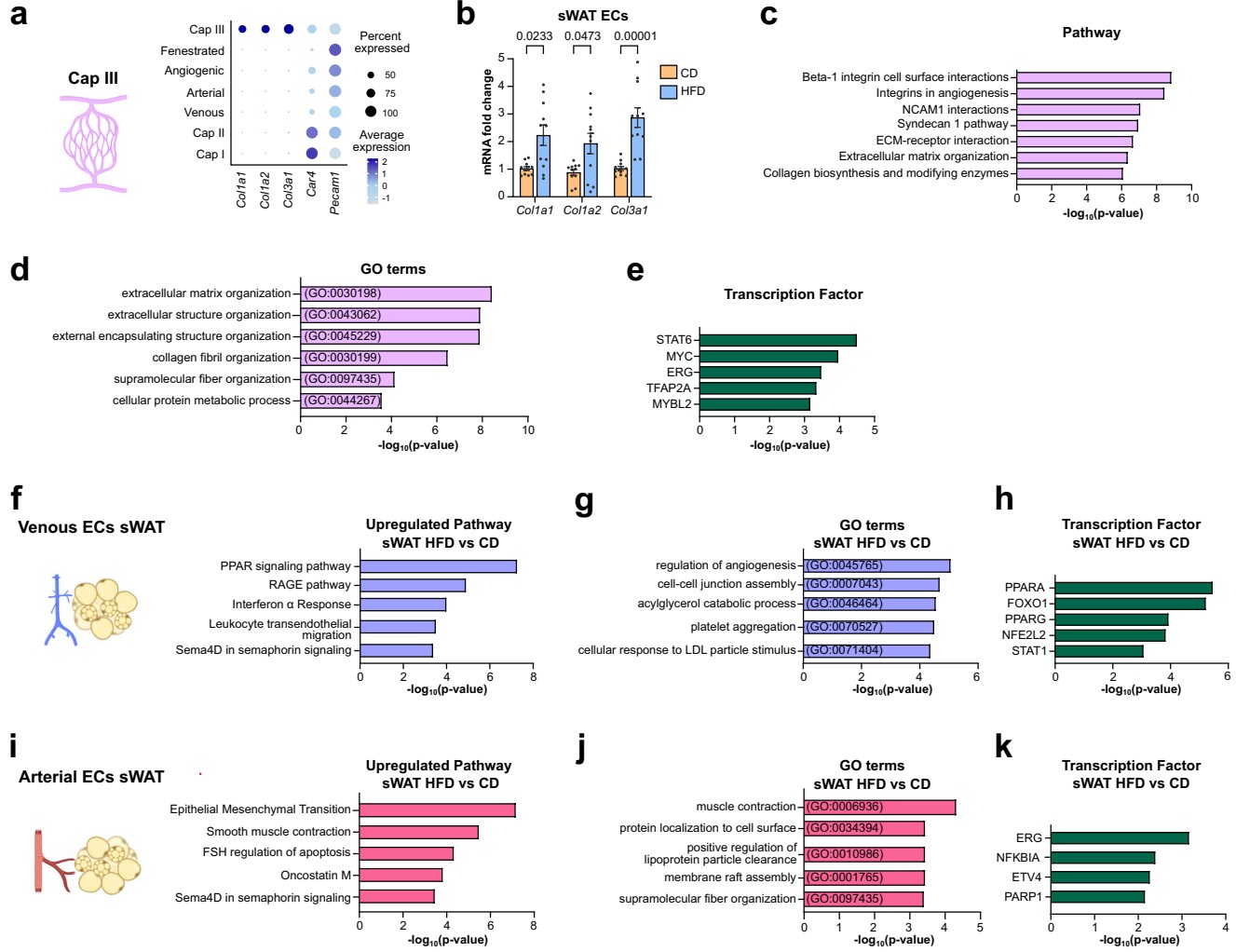

**Fig. 3 | Obesity driven vascular alterations in sWAT. a** Dot plot depicting enriched marker genes in Cap III endothelial cells (ECs). **b** Quantitative real-time PCR detection of collagen genes in subcutaneous white adipose tissue (sWAT) ECs isolated from lean (control diet, CD n = 11 mice) and obese (high fat diet, HFD n = 11 mice) animals. **c** Bioplanet annotated upregulated pathways in Cap III ECs. **d** Gene ontology (GO) terms enriched in Cap III ECs. **e** TTRUST transcription factor (TF) analysis of upstream regulators in Cap III ECs. **f** Upregulated pathways **g** enriched

GO terms and **h** predicted TFs in venous ECs in obese (high fat diet, HFD) vs lean (control diet, CD) sWAT. **i** Upregulated pathways **j** enriched GO terms and **k** predicted TFs in arterial ECs in obese (HFD) vs lean (CD) sWAT. Data represents ± SEM and two-sided Mann-Whitney test (**b**), Fisher's exact test (**c–k**). All schematics created in BioRender. Hasan, S. (2025) https://BioRender.com/0eux7g4. Source data are provided as a Source Data file.

receptor transcripts in comparison to other EC subtypes (Fig. 4i–k). Bulk mRNA analysis of freshly isolated sWAT ECs revealed a reduction of *Vegfr* transcript levels in obese tissue (Fig. 4l). We also detected reduced *Kdr* (*Vegfr2*) transcripts in vWAT ECs (Supplementary Fig. 7c–f).

Paracrine VEGF signaling is pivotal in the formation and maintenance of fenestrated endothelium[20]. Besides paracrine signaling, an autocrine VEGFA signaling is also crucial for maintaining a healthy endothelium[45]. While no significant changes were observed in endothelial *Vegfa* transcript in vWAT ECs (Supplementary Fig. 7g), endothelial *Vegfa* transcripts were significantly reduced in sWAT of obese mice (Fig. 4m), which could also explain the observed vascular rarefaction.

### Cellular source of VEGFA expression in WAT and changes in obesity

VEGF has been extensively studied in the context of angiogenesis and vascular remodeling in adipose tissue[46]. Nevertheless, role of VEGFA in forming and maintaining fenestrated endothelium and sustaining EC

survival in WAT remains poorly understood. To gain understanding of VEGFA expression patterns in the context of obesity, we performed immunostaining for VEGFA in sWAT. VEGFA expression could be detected around larger vessels (arrowheads, Fig. 5a) and some dispersed expression in the capillary beds (Fig. 5b). VEGFA expression was substantially reduced in obese sWAT around large caliber vessels (Fig. 5c). However, we did not observe any difference in the capillary area (Fig. 5d). Also, no notable difference in VEGFA expression was observed in vWAT (Supplementary Fig. 7h–k).

Since paracrine VEGF signaling is essential for formation and maintenance of fenestrated endothelium[20], we postulated that VEGFA+ cells might represent a crucial source of this signaling. To ascertain the localization of VEGFA+ cells with respect to the endothelium, we performed whole mount staining for VEGFA together with PLVAP (Fig. 5e, f). To determine the number of VEGFA+ cells in proximity to PLVAP+ blood vessels, we employed surface rendering around PLVAP+ vessels with distance transformation. This approach allowed us to distinguish between cells in close proximity (magenta spheres, blue arrowheads) and those at a distance (green spheres,

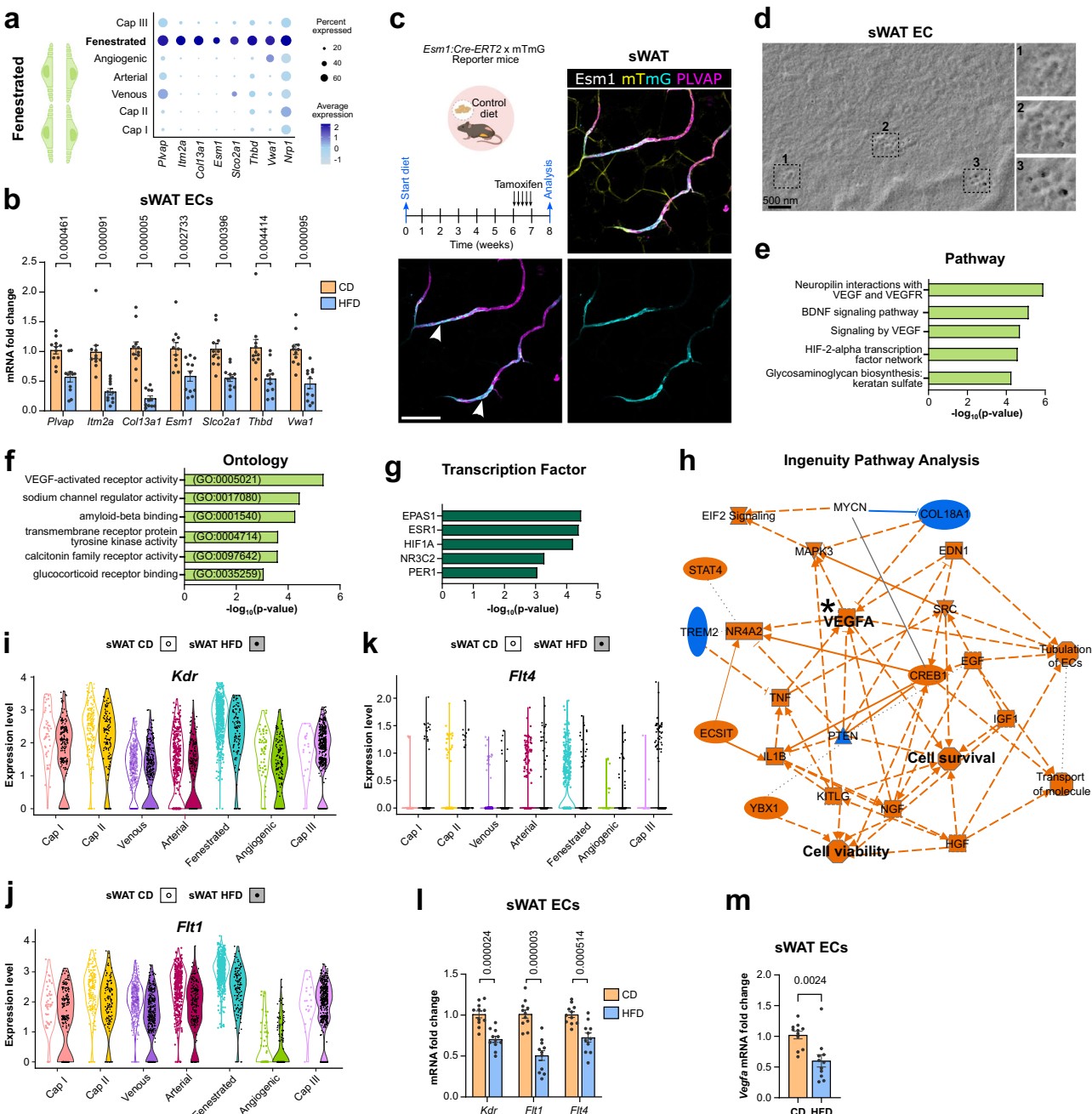

**Fig. 4 | Fenestrated endothelium in sWAT. a** Dot plot depicting enriched marker genes in fenestrated endothelial cells (ECs). **b** Quantitative real-time PCR detection of marker genes of fenestration in subcutaneous white adipose tissue (sWAT) ECs isolated from lean (control diet, CD n = 11 mice) and obese (high fat diet, HFD n = 11 mice) animals. **c** Representative orthogonal projections from whole mount staining for PLVAP (magenta) performed on sWAT from Esm1:Cre-ERT2 x mTmG reporter mice. Upon tamoxifen treatment, the reporter switches from tdtomato (yellow) to GFP (cyan). GFP+ blood vessels are also PLVAP+ (arrowheads). **d** Scanning electron micrograph showing fenestrae (see insets) in sWAT ECs in vitro. **e** Bioplanet annotated upregulated pathways in fenestrated ECs. **f** Gene ontology (GO) terms enriched in fenestrated ECs. **g** TTRUST transcription factor analysis (TF) of upstream regulators in fenestrated ECs. **h** Ingenuity pathway analysis (IPA) on differentially expressed marker genes in this cluster. Asterisk denotes VEGFA signaling. Orange and blue graphics represent predicted activation and inhibition, respectively. Straight lines and dotted lines represent direct or indirect interaction, respectively. **i**–**k** Violin plot depicting *Vegf* receptor levels in sWAT ECs in lean (CD) and obese (HFD) mice. **l** Quantitative real-time PCR detection of *Vegf* receptors and **m** *Vegfa* in sWAT ECs isolated from lean (CD n = 11 mice) and obese (HFD n = 11 mice) animals. Scale bar 100 μm. Data represents ± SEM, two-sided Welch's t-test (**b**, **l**), Fisher's exact test (**e**–**g**), two-sided Mann-Whitney test (**m**). Schematics created in BioRender. Hasan, S. (2025) https://BioRender.com/7rbmeru. Source data are provided as a Source Data file.

white arrowheads) from PLVAP+ vessels (Fig. 5g, h). A notable decline in VEGFA+ cells in close proximity to PLVAP+ vessels was observed in obese sWAT (Fig. 5i). Instead, a considerable number of VEGFA+ cells were located in the crown-like structures associated with dying adipocytes in obese sWAT (white arrowheads, Fig. 5f, h). In addition, we observed a decline in PLVAP+ vessels in obese sWAT (Fig. 5j). To

elucidate the identity of VEGFA+ cells in the vascular niche, we reanalyzed publically available single-cell transcriptomics data (Fig. 5k)[47]. Our findings revealed that *Vegfa* expression is predominantly restricted to adipocytes, with some expression also in adipocyte stem and progenitor cells (ASPCs) (Fig. 5l), which are known to occupy a vascular niche[48].

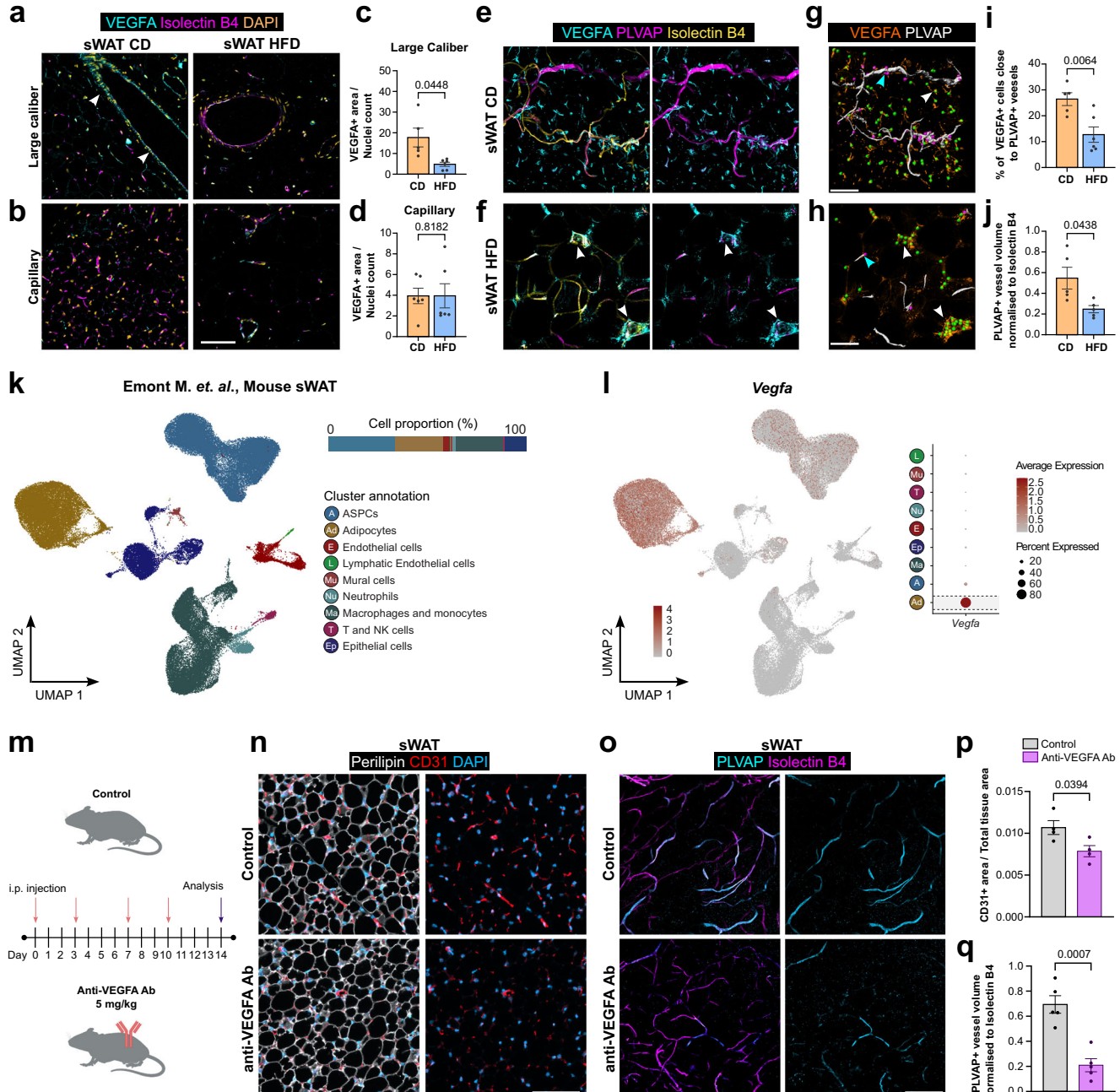

**Fig. 5 | VEGFA expression in adipose tissue and effect of systemic VEGFA blockage on fenestrated endothelium. a** Representative confocal images of VEGFA (cyan) and Isolectin B4 (magenta) staining in subcutaneous white adipose tissue (sWAT) sections around large vessels (arrowheads) or **b** capillary beds in lean (control diet, CD) and obese (high fat diet, HFD) mice. **c** Quantification of VEGFA+ area around large vessels (CD *n* = 5, HFD = 6 mice) or **d** capillary beds (CD *n* = 6, HFD = 6 mice). **e** Representative orthogonal projections from whole mount staining for VEGFA (cyan), PLVAP (magenta), and isolectin B4 (yellow) in sWAT in lean and **f** obese mice. **g** Volumetric reconstruction of PLVAP+ blood vessels showing relative distance of VEGFA+ cells to rendered blood vessel in lean and **h** obese mice. VEGFA+ cells close to (<10 μm, magenta spheres, blue arrowhead) or away from (>10 μm, green spheres, white arrowhead) traced blood vessel are depicted. **i** Quantification of VEGFA+ cells close to PLVAP+ blood vessels in lean (*n* = 5 mice) and obese (*n* = 6 mice) animals. **j** Quantification of PLVAP+ blood vessel in lean (*n* = 5 mice) and obese (*n* = 5 mice) animals. **k** Uniform manifold approximation and

projection (UMAP) of clusters formed by 91,097 cells of murine sWAT generated by Emont M. et al., (single nucleus RNA sequencing) depicting cell proportion and annotation. **l** Feature plot and dot plot showing expression of *Vegfa* in murine dataset. **m** Schematic representation of control or anti-VEGFA antibody (Ab) treatment regimen. **n** Representative confocal images showing CD31+ (red) blood vessels, Perilipin+ (white) adipocyte and DAPI+ (cyan) nuclei in sWAT sections from mice injected with control or anti-VEGFA Ab. **o** Representative orthogonal projections from whole mount staining for PLVAP (cyan) and isolectin B4 (magenta) in sWAT from mice injected with control or anti-VEGFA Ab. **p** Quantification of blood vessel (CD31 + ) area in (**n**) (*n* = 4 mice each). **q** Quantification of PLVAP+ blood vessel volume in (**o**) (*n* = 5 mice each). Scale bars 100 μm. Data represents ± SEM, two-sided Mann-Whitney test (**d**) and two-sided Welch's t-test (**c, i, j, p, q**). Schematics created in BioRender. Hasan, S. (2025) https://BioRender.com/k8kxui3. Source data are provided as a Source Data file.

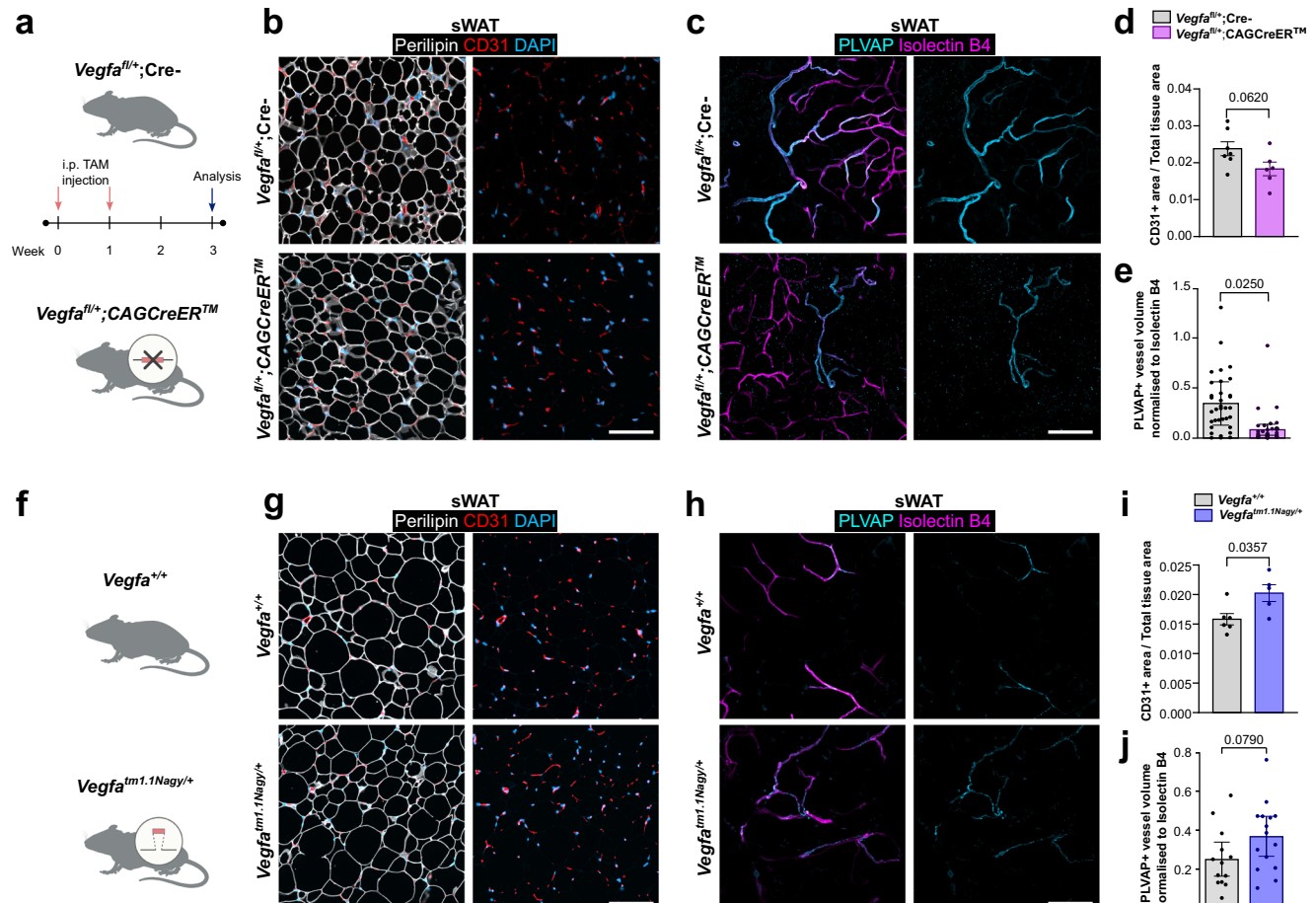

**Fig. 6 | Effect of genetic loss and gain of *Vegfa* function on fenestrated endothelium. a** Schematic representation of tamoxifen (TAM) treatment for *Vegfa* loss-of-function mice. **b** Representative confocal images showing CD31+ (red) blood vessels, Perilipin+ (white) adipocyte and DAPI+ (cyan) nuclei in subcutaneous white adipose tissue (sWAT) sections from V*egfa*^fl/+ mice lacking Cre (*n* = 7 mice) and *Vegfa*^fl/+;CAGCreER^TM mice (*n* = 6 mice). **c** Representative orthogonal projections from whole mount staining for PLVAP (cyan) and isolectin B4 (magenta) in sWAT from *Vegfa*^fl/+ mice lacking Cre (*n* = 6 mice) and *Vegfa*^fl/+;CAGCreER^TM mice (*n* = 6 mice). **d** Quantification of blood vessel (CD31+) area in (**b**). **e** Quantification of PLVAP+ blood vessel normalized to isolectin B4 in (**c**). Each dot represents one imaged area (control-34 maximum intensity projections analyzed from 6 mice, *Vegfa* loss-of-function-36 maximum intensity projections analysed from 6 mice). **f** Schematic representation of *Vegfa* gain-of-function mice. **g** Representative

confocal images showing CD31+ (red) blood vessels, Perilipin+ (white) adipocyte and DAPI+ (cyan) nuclei in sWAT sections from *Vegfa*^+/+ mice (*n* = 6 mice) and *Vegfa*^tm1.1Nagy/+ mice (*n* = 5 mice). **h** Representative orthogonal projections from whole mount staining for PLVAP (cyan) and isolectin B4 (magenta) in sWAT from *Vegfa*^+/+ mice (*n* = 4 mice) and *Vegfa*^tm1.1Nagy/+ mice (*n* = 4 mice). **i** Quantification of blood vessel (CD31+) area in (**g**). **j** Quantification of PLVAP+ blood vessel normalized to isolectin B4 in (**h**). Each dot represents one imaged area (control-13 maximum intensity projections analyzed from 4 mice, *Vegfa* gain-of-function-15 maximum intensity projections analysed from 4 mice). Scale bars 100 μm. Data represents ± SEM, two-sided Welch's t-test (**d, i**), linear regression with cluster-robust standard errors (**e, j**). Schematics created in BioRender. Hasan, S. (2025) https://BioRender.com/e743pdb. Source data are provided as a Source Data file.

## VEGFA helps to maintain fenestrated endothelium in sWAT

To elucidate the role of VEGF signaling in sWAT fenestrated endothelium, we performed a VEGF blockade experiment where we treated mice with anti-VEGFA monoclonal antibody (B20-4.1) (Fig. 5m) and analyzed its impact on vascular density and fenestrated endothelium. Immunohistological staining on gross sWAT section revealed a reduction in vascular density in mice treated with anti-VEGFA (Fig. 5n, p). Furthermore, analysis of whole-mount staining also showed a decline in PLVAP+ vessels in anti-VEGFA treated mice compared to controls (Fig. 5o, q).

To complement the systemic VEGFA blockade, we also performed genetic *Vegfa* loss-of-function experiments in mice (Fig. 6a). Quantitative immunohistological analysis showed that *Vegfa*^fl/+;CAGCreER^TM mice had similar vascular density in the sWAT when compared to *Vegfa*^fl/+ littermates lacking Cre (Fig. 6b, d). However, a trend towards lower vascular density was observed. Whole-mount staining of sWAT revealed a decrease in PLVAP+ blood vessels in *Vegfa*^fl/+;CAGCreER^TM mice compared to *Vegfa*^fl/+ littermates lacking Cre (Fig. 6c, e).

In order to ascertain the impact of genetic supplementation of *Vegfa* on sWAT vasculature, we conducted *Vegfa* gain-of-function experiments in mice (Fig. 6f). Quantitative immunohistological analysis revealed that *Vegfa*^tm1.1Nagy/+ mice had increased vascular density in sWAT when compared to *Vegfa*^+/+ littermates (Fig. 6g, i). Whole-mount staining of sWAT revealed that *Vegfa*^tm1.1Nagy/+ mice did not exhibit significant alterations in PLVAP+ vessels when compared to littermate controls although a trend towards an increase in PLVAP+ blood vessels could be observed (Fig. 6h, j). In conclusion, these findings suggest that VEGFA helps to maintain fenestrated endothelium in sWAT.

## Differential endothelial response during short term high-fat diet

Next, we wanted to investigate whether short-term high-fat diet (four weeks) would induce comparable alteration in the endothelium as with long-term feeding. Analysis of vascular density did not exhibit any vascular rarefaction in sWAT at this stage (Supplementary Fig. 8a, b). Additionally, no discernible difference was observed in the transcript levels of markers associated with fenestrated endothelium

(Supplementary Fig. 8c). Furthermore, *Vegfa* transcript levels in the endothelium remained unaltered (Supplementary Fig. 8d). The levels of *Kdr* were slightly reduced in ECs of sWAT after four weeks of HFD (Supplementary Fig. 8e). Conversely, VEGFA expression showed an elevated tendency in both large-caliber vessels and capillaries within obese sWAT (Supplementary Fig. 8f–i).

Similarly, we did not observe a difference in vascular density in vWAT (Supplementary Fig. 8j, k). However, we did detect a notable increase in the *Esm1* transcript (Supplementary Fig. 8l). VEGFA expression was significantly elevated in areas of large vessels and a tendency to be elevated in capillaries (Supplementary Fig. 8m-p) in obese vWAT, but endothelial *Vegfa* transcripts did not change (Supplementary Fig. 8q). In addition, *Vegfr* transcripts were elevated in obese vWAT ECs (Supplementary Fig. 8r).

Since we observed a differential response in four weeks vs eight weeks of HFD, we also characterized WAT vasculature and VEGFA expression at an intermediate time point of six weeks HFD (Supplementary Fig. 9a–r). Most of the parameters exhibited a downward trend, comparable to that observed at eight weeks HFD, thereby reinforcing the notion that there is an initial increase in VEGFA expression with short-term high-fat diet, which subsequently diminishes over time.

To elucidate the alterations in the WAT profile occurring between four and eight weeks of HFD, we performed an adipokine array on sWAT tissue lysate at different time points (Supplementary Fig. 9s). Several parameters including Lipocalin-2, TIMP-1, Pentraxin-2, which have been associated with tissue inflammation and remodeling[49,50], exhibited a differential response between four weeks and eight weeks of HFD (Supplementary Fig. 9t).

Taken together, obesity induces vascular alterations in a depot-specific manner, with a differential response of VEGFA in early and later stages of tissue expansion.

### Fenestrated endothelium in human sWAT

To gain insight into the expression patterns of EC populations that may indicate signatures of fenestrated ECs, we explored a recently published dataset for human sWAT[51]. In this dataset, *PLVAP* expression was reported in both venous and capillary EC populations[51]. Interestingly, the expression was restricted to a subpopulation of each vessel type, suggesting that only a subset of specialized vessels may display fenestrated characteristics. To validate these findings, we performed immunohistochemical staining for ITM2A in sWAT from lean donors and patients with obesity (Fig. 7a). ITM2A expression was significantly reduced in sWAT from patients with obesity compared to lean sWAT (Fig. 7c). Additionally, we also detected ITM2A+ ECs in human vWAT (Fig. 7b), which are also reduced in vWAT from patients with obesity (Fig. 7d).

## Discussion

This study provides comprehensive insights into the impact of obesity on adipose tissue-derived ECs in a depot-specific manner. By combining single-cell transcriptomics with the Cell Hashing technique, we generated a detailed landscape of endothelial heterogeneity and vascular alterations associated with obesity. The Cell Hashing technique enabled us to perform more robust statistical analyses, detect doublets with greater accuracy, and mitigate batch effects[13].

Our initial classification resulted in identification of seven EC subtypes in WAT, which were observed at distinct frequencies in lean and obese mice in a depot-specific manner. We could demarcate key differences in the endothelial response to obesity in both macro and microvasculature in a depot-specific manner. While the sWAT endothelium elicits a more inflammatory response, the vWAT vasculature activates a vascular augmentation program across several vascular beds. Multiple studies have reported that the expansion of WAT results in hypoxia which in turn activates HIF-1α signaling, thereby initiating

an angiogenic response. Nevertheless, the development of new blood vessels does not necessarily guarantee the functionality and perfusion of the resulting vasculature[52]. Additionally, HIF-1α signaling has been linked to extensive ECM remodeling and induction of fibrosis in vWAT, leading to vWAT dysfunction[28]. Furthermore, we identified an EndMT signature in obese vascular beds. Endothelial populations showing signs of EndMT have been reported in both porcine WAT[53] and in WAT of patients with obesity[54]. In a pathological setting, EndMT is predominantly triggered by inflammatory mediators, resulting in the loss of endothelial identity and eventually endothelial dysfunction[55].

Cap III ECs, which are present in obese sWAT, show an upregulation for processes involved in collagen biosynthesis, ECM organizations and integrin interactions during angiogenesis. This signature exhibits congruency with a dataset derived from human sWAT during obesity[47,56]. We speculate that this population may be involved in pathological angiogenesis, thereby aggravating fibrosis in sWAT during obesity, as a similar *Col1a1* expressing EC population was identified driving pathological angiogenesis during diabetic retinopathy[57].

One of the most remarkable EC subtypes that we detected in sWAT is the fenestrated endothelium. VEGF signaling is important for maintaining fenestrations in several vascular beds[58,59]. PLVAP is a crucial structural component of fenestrated endothelia and there are reports suggesting its regulation by VEGFA in physiological and pathological settings[60–62]. However, the link between VEGF signaling and endothelial fenestrations in WAT has not been studied before. Utilizing systemic anti-VEGFA blockade and genetic *Vegfa* manipulation, we could demonstrate that VEGFA is necessary for maintaining fenestration in sWAT. However, the biological significance of this population in adipose health and disease remains uncertain. Several endocrine organs contain fenestrated capillaries, which facilitate efficient exchange of nutrients, hormones, and small peptides. A recent study has also underscored the role of fenestrae in liver sinusoidal ECs where they facilitate very low-density lipoprotein (VLDL) transport. During obesity, the upregulation of SEMA3A leads to defenestration, which ultimately results in hepatic steatosis[63]. Given that WAT is both endocrine and a metabolic organ, it seems reasonable to posit that these fenestrations aid in transport. However, the secretome profiles of sWAT and vWAT were nearly identical, and fenestrated ECs were found exclusively in a subset of sWAT ECs. In light of the in silico data and whole-mount staining it seems plausible to suggest that the VEGFA + cells could be ASPCs that are known to occupy a vascular niche. Endothelial fenestrations may facilitate the exchange of instructive and regulatory signals between both cell types[48]. However, this hypothesis warrants further research. Nevertheless, the broader physiological role of this cell type in adipose tissue homeostasis is still uncertain and this remains a limitation of this study.

The role of VEGF signaling in adipose tissue biology has been extensively studied albeit with contradicting evidence regarding levels of VEGFA in WAT during obesity[3]. In our own study, we encounter depot-specific temporal variations in the levels of VEGFA in WAT, in response to high-fat diet. A short term high-fat diet resulted in higher levels of VEGFA and maintenance of vascular density, while a longer feeding resulted in reduction of VEGFA levels and a corresponding decreased in vascular density. These results may account for the discrepancy previously observed in the effect of VEGFA ablation as a therapeutic measure for obesity and associated metabolic syndromes. Taken together, these observations point towards a complex dichotomous response of VEGFA and its effect on angiogenesis and vascular maintenance in the context of obesity depending upon the stage of WAT expansion.

A similar single-cell transcriptomics profiling was performed by Bondareva et al.[10] to study the impact of obesity on WAT ECs. We found several congruencies between our datasets, including induction of focal adhesion, ECM organization pathways, integrin signaling, and hypoxia induced signaling in obese WAT ECs. We also observed a

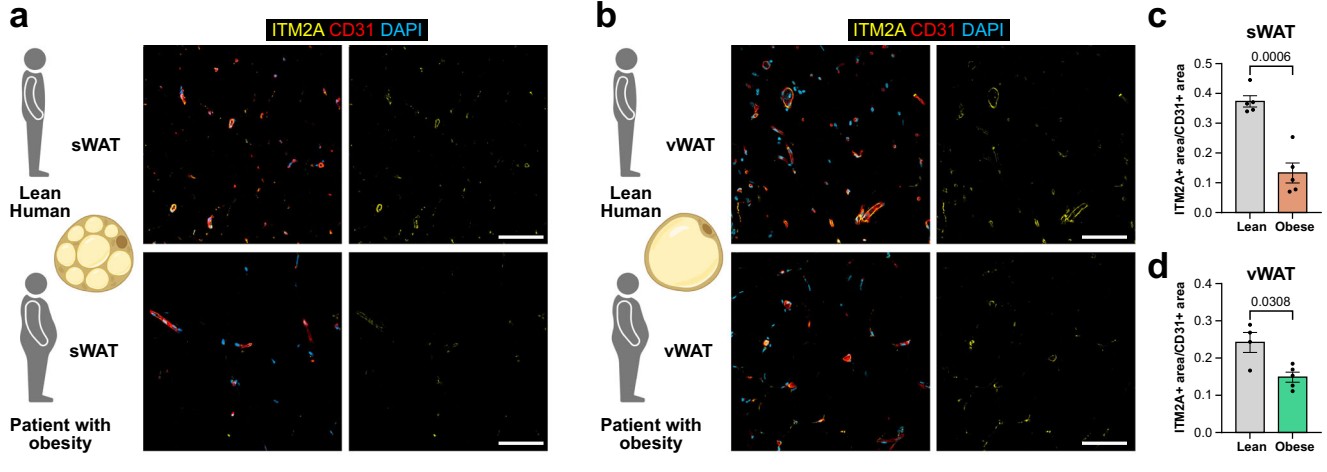

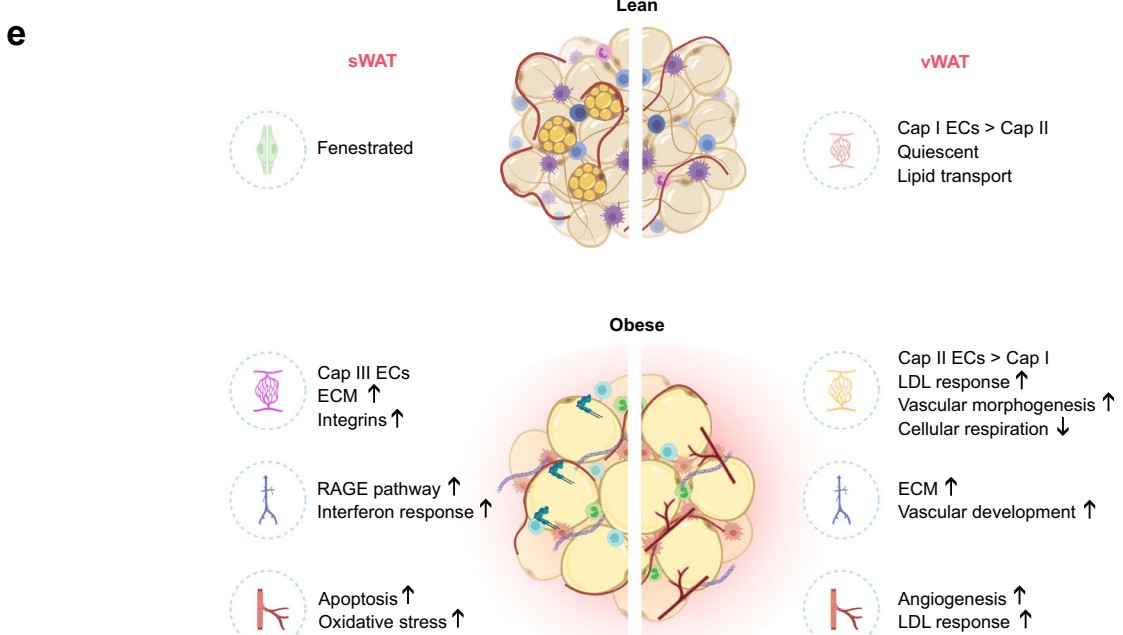

**Fig. 7 | Fenestrated endothelium in human sWAT. a** Representative confocal images showing ITM2A+ (yellow) endothelial cells (ECs) and CD31+ (red) blood vessels in human subcutaneous white adipose tissue (sWAT) from lean human or patients with obesity. **b** Representative confocal images showing ITM2A+ (yellow) ECs and CD31+ (red) blood vessels in human visceral white adipose tissue (vWAT) from lean human or patients with obesity. **c** Quantification of ITM2A+ area normalized to CD31+ area for sWAT (lean donors *n* = 5, patient with obesity *n* = 5).
**d** Quantification of ITM2A+ area normalized to CD31+ area for vWAT (lean donors *n* = 4, patient with obesity *n* = 5. Schematics created in BioRender. Hasan, S. (2025) https://BioRender.com/4t8j1lu. **e** Graphical summary of obesity-associated vascular alterations in sWAT and vWAT ECs. Graphical summary created in BioRender. Hasan, S. (2025) https://BioRender.com/5ghaxuk. Scale bars 100 µm. Data represents ± SEM, two-sided Welch's t-test. Source data are provided as a Source Data file.

downregulation of cellular respiration in obese vWAT capillaries. On the other hand, we do observe a trend towards increased angiogenic ECs during obesity; however, we did not detect any significant proliferating ECs in our dataset. Similar to our study, Bondareva et al. also reported the presence of fenestrated ECs subtype exclusively in sWAT, which is reduced under obese conditions.

Our study goes further ahead and focuses in detail on the biology of fenestrated endothelium in adipose tissue, validating their presence using modalities like reporter mice and whole mount staining as well as visualizing the presence of fenestrae at an ultrastructural level. Additionally, utilizing systemic approaches as well as genetic mouse models we could establish the significance of VEGF signaling in maintaining fenestrated endothelium in sWAT. We also included several time

points of high-fat feeding to highlight the temporal differences in VEGFA response. Additionally, we went one-step further to validate this cell population in human WAT and could show that they are reduced in patients with obesity. Unlike murine WAT where fenestrated endothelium is limited to sWAT, we detected ITM2A+ ECs not only in subcutaneous but also in visceral fat compartments in human donors. It is important to understand that several differences exist between mice and human WAT in terms of location, vascularization, adipocyte size, and immune cell milieu[64]. Visceral WAT in mice is mostly perigonadal, whereas the most accessible and studied visceral WAT in human is omental WAT. Human vWAT is more vascularized with smaller adipocytes compared to human sWAT[65], while the opposite is true for mice. Therefore, given the existing biological

differences it is not surprising that we observe different patterns in terms of fenestrated ECs between mice and human.

In conclusion, our study provides a comprehensive transcriptome resource of depot-specific endothelial heterogeneity of WAT under lean and obese conditions. Our comprehensive characterization of the distinct endothelial populations provides key insights into the mechanisms underlying obesity-associated endothelial dysfunction (Fig. 7e) and provides a foundation for future research to pave the way for novel therapeutic intervention.

## Methods

### Animal models
All animal procedures were performed in accordance with institutional and national regulations and approved by local committees for animal experimentation (G226/20 RP Karlsruhe, DKFZ; PP1810839 UK Home Office) Animals were housed under specific pathogen-free barrier conditions at $21 \pm 2\,°C$ with 60 % humidity and a 12 h light /dark rhythm and provided with chow diet (unless specified) and water ad libitum. Wild type C57BL/6J mice were purchased from Janvier Labs. Animals were euthanized by cervical dislocation.

**Diet-induced obesity.** C57BL/6J male mice were put on a high fat diet containing 60% fat, 20% carbohydrate, 20% proteins (Cat# D12492i, Research Diets) or a control diet with 10% fat, 70% carbohydrate, 20% proteins (Cat# D12450Bi, Research Diets), starting from 6 to 8 weeks of age for indicated amount of time.

**Esm1:Cre-ERT2 × Rosa26-mtdtomato-mEGFP (mTmG) reporter mice.** Heterozygous Esm1-CreERT2[66] Cre+ x homozygous Rosa26 mTmG male mice were put on a control diet starting from 6 to 8 weeks of age. After 6 weeks of diet, mice were treated with tamoxifen (100 mg/kg body weight) in peanut oil by oral gavage for 5 days consecutively. After 8 weeks of diet in total mice were sacrificed to collect organs.

**Anti-VEGFA Antibody treatment.** 12-week-old C57BL/6 J male mice were injected i.p. with either control or anti-VEGFA antibody (B20-4.1, 5 mg/kg) twice a week for two weeks (4 doses in total). Mouse were sacrificed to harvest organs two weeks from the first dose.

**Vegfa loss-of-function.** The study was performed following UK Home Office and institutional Animal Welfare and Ethical Review Body (AWERB) guidelines. All mice were bred at the UCL Institute of Ophthalmology. Mice carrying the *Vegfa*$^{fl}$ allele[67] were maintained on a mixed genetic background (C57BL6/J and 129/Sv) and backcrossed to C57BL6/J wild-type mice (Charles River Laboratories) carrying a tamoxifen-inducible *Cre* transgene under the control of the ubiquitous chicken actin promoter (CAG-CreER™[68]). Cre-mediated recombination was induced in adult male *Vegfa*$^{fl/+}$;CAGCreER™ mice by intraperitoneal (i.p.) injections of 0.5 mg tamoxifen (6 mg/mL in corn oil, Sigma-Aldrich), once a week for 2 weeks. Age-matched tamoxifen-injected *Vegfa*$^{fl/+}$ lacking Cre mice were used as control group. sWAT was harvested 2 weeks after the last i.p. injection for analysis.

**Vegfa gain-of-function.** The study was performed following UK Home Office and institutional Animal Welfare and Ethical Review Body (AWERB) guidelines. All mice were bred at the UCL Institute of Ophthalmology. *Vegfa*$^{tm1.1Nagy}$ mice[69] on a CD1 background were backcrossed to C57BL/6J wild-type mice (Charles River Laboratories). sWAT was collected from 14-week-old male mice maintained on a standard diet.

### Human adipose tissue collection
The collection of visceral and subcutaneous fat tissue samples was performed according to the Declaration of Helsinki and approved by the local ethics committee (approval #38/4/21, University Medical Center Göttingen, Germany). Patients with obesity were considered to have a BMI > 40 kg/m2, while lean controls were considered to have a BMI < 25 kg/m2. Tissue from male human donors was used in this study. Inclusion criteria included age ≥18–60 years and the ability to provide informed consent. General exclusion criteria were pregnancy, inflammatory bowel disease, severe cardiovascular disease (NYHA class III or IV), or cancer. Written informed consent was obtained by all the participants of this study and fat samples (10–50 g) were taken during a scheduled surgery for obesity or in case of the lean controls e.g., hernia repair. Half of each sample was stored fresh-frozen at −80 °C and the other half was embedded after fixation in formalin.

### Sample preparation for single cell RNA sequencing
**Single cell suspension.** Inguinal fat pad representing the subcutaneous white adipose tissue (sWAT) and epididymal fat pad representing the visceral WAT (vWAT) were dissected from lean and obese mice. Prior to mincing the tissue, lymph nodes were removed from sWAT to avoid any contamination from lymphoid structures. Minced tissue was digested with collagenase II (sWAT 2 mg/ml, vWAT 1 mg/ml; Gibco) and dispase II (sWAT 2 mg/ml, vWAT 1 mg/ml; Sigma) in PBS containing 1% fatty acid free bovine serum albumin (FA-BSA) at 37 °C for 45–60 min with rigorous shaking every 10 mins. The digestion was quenched with PBS containing 1% BSA and 5 mM EDTA (FACS buffer) and the homogenate was filtered through a 100 μm cell strainer and centrifuged at $300 \times g$ for 5 min. The supernatant was aspirated and pellet was resuspended in RBC lysis buffer for 1 min. Cells were rinsed in FACS buffer and filtered through a 40 μm filter, and resuspended in 1 ml FACS buffer. Cells were counted to optimize antibody concentration in the next step.

**Cell Hashing and FACS.** Cell Hashing was performed according to manufacturer's recommendation. The following Hashtag antibodies were used. TotalSeq™- A0301 anti-mouse Hashtag 1 antibody (Biolegend Cat# 155801), TotalSeq™- A0302 anti-mouse Hashtag 2 antibody (Biolegend Cat# 155803), TotalSeq™- A0303 anti-mouse Hashtag 3 antibody (Biolegend Cat# 155805) to label cells from each biological replicate ($n = 3$). Cell Hashing antibody staining was performed simultaneously in the same step as the staining for FACS. The following FACS antibodies were used; CD45-FITC (BD Cat# 553079; 1:200), CD31-APC (BD Cat# 561814; 1:100). Cells were incubated in the antibody cocktail for 30 min on ice and incubated for 5 min with DAPI. Cells were washed 3 times and resuspended in FACS buffer without EDTA. At this stage cells derived from biological replicates were pooled. Cells were sorted with BD FACS Aria cell sorter equipped with a 100 μm nozzle selecting for live single cells All compensation was performed using compensation beads for single color staining and cells for negative and DAPI staining. Cells were gated to collect CD31 + CD45- population after excluding debris, dead cells (70–80% viable cells) and doublets.

**Single-cell workflow.** Single-cell RNA sequencing was performed using a 10X Genomics Chromium Next GEM Single Cell 3' GEM Kit v3.1 (Cat # 1000128) with a Next GEM Chip G (Cat # 1000127) using the manufacturer's protocol. In addition to preparing the mRNA library, the hashtag library was also generated according to manufacturer's protocol (Biolegend). Briefly, at the cDNA amplification step additional primers were added to amplify the Hashtags. After cDNA amplification, Hashtag library (180 bp) and mRNA derived library (>300 bp) were separated and amplified. All the primers and indices used for Hashtag library preparation are mentioned in Supplementary Table 1. The libraries were multiplexed and mixed at the ratio of 1:12 (Hashtag library: mRNA derived cDNA library) and sequenced on NovaSeq 6000.

## Bioinformatics analysis

**Data processing.** Raw reads were mapped against the mouse genome (GRCm38.97) via *Starsolo* (version 2.7.9) with default parameters[70]. Additionally, the barcodes were counted via the *CITE-seq-Count* package (version 1.4.3) with the parameters *-cbf 1 -cbl 16 -umif 17 -umil 28 -cells 40000* (10.5281/zenodo.2590196). Both count matrices were imported with Seurat's *Read10X* function and analyzed following the commands from the "Demultiplexing with hashtag oligos" vignette from Seurat3 (https://satijalab.org/seurat/articles/hashingvignette.html). UMI- and barcode-matrix for each condition were normalized and scaled via log- normalization and centered-log-normalization, respectively. Individual cite-seq barcode matrix was subsetted based on the cell names from the single cell matrix. Afterwards both matrices were joined based on common cell names. To assign single cells back to their sample origins, the HTODemux function was used with a positive.quantile = 0.99. Barcode doublet and negative cells were removed. These steps were performed individually for each condition (SWAT.HFD, SWAT.CD, EWAT.HFD, and EWAT.CD). The individual sample were merged into a common Seurat object. DoubletFinder (version 2.0.3) was applied to remove potential cross-sample doublets. Variable feature detection was performed with "mean.var.plot" as selection method. For clustering, the resolution was set to 0.6, followed by UMAP clustering on the first 10 principal components.

**Identification of differentially expressed genes (DEGs).** "FindAll-Markers" function was implemented in Seurat to identify marker genes for each cluster with the options "min.pct = 0.25, logfc.threshold = 0.25" and selected with adjusted *p*-value < 0.05. For identifying DEGs between two clusters or conditions, "*FindMarkers*" function was used utilizing the Wilcoxon-rank-sum test and identical parameters as above.

**Pathway enrichment and transcription factor analysis.** All pathway enrichment and transcription factor analysis was performed with EnrichR[71], Top 100 upregulated genes were used for analysis from DEGs list. Bioplanet annotated pathway and KEGG pathway databases were utilized. For TF analysis TTRUST database was used. Ingenuity pathway analysis (IPA) was performed using IPA software (Qiagen)[72]. The entire gene list generated from DEGs analysis was used for IPA.

## Whole mount immunofluorescence

Lean and obese mice were anesthetized and intracardially perfused with PBS and 1% paraformaldehyde (PFA) (pH 7.4). WAT was dissected and further cut into 2 mm × 2 mm pieces and transferred to 1% PFA for further fixation at room temperature (RT) for 1 h. Tissue samples were washed with PBST (0.3% Tween-20), 5 times for 5 min each. Tissue samples were blocked in 10% goat or donkey serum in PBST for 1 h at RT. Samples were incubated in the following primary antibody diluted in blocking solution for 16 h at 4 °C. PLVAP (1:50, MECA-32c, DHSB); VEGFA (1:100, ab52917, EP1176Y, Abcam); GJA5 (1:50, ADI-CX40-A, Alpha Diagnostics); P-Selectin (1:25, 553742, BD Biosciences); CAR4 (1:50, AF2414-SP, R&D). Samples were washed three times in PBST and incubated in the following secondary antibodies diluted (1:200) in PBST for 1 h at RT. Goat anti-rat 546 (Invitrogen, A-11081), Goat anti-rat 647 (Invitrogen, A-21247), Goat anti-rabbit 488 (Invitrogen, A-11034), Goat anti-rabbit 546 (Invitrogen, A-11035), Goat anti-rabbit 647 (Invitrogen, A-21245), Donkey anti-goat 546 (Invitrogen, A-11056), Donkey anti-goat 647 (Invitrogen, A-21447), Donkey anti-rat 488 (Invitrogen, A-21208), Donkey anti-rabbit 647 (Invitrogen, A-31573), Goat anti-mouse 546 (Invitrogen, A-11030). In addition, Isolectin B4-568 (Invitrogen I21412) or 647 (Invitrogen I32450) was added along with the secondary antibodies when indicated. Samples were counter stained with DAPI for 5 min and then washed three times with PBST for 5 min each followed by two washes with PBS. Samples were trimmed and mounted in fluorescence mounting medium (Agilent Dako, S3023) on cavity slides (BRAND, 475535) and imaged on a confocal microscope (LSM 710, Carl Zeiss).

## Immunohistochemistry

Lean and obese mice were anesthetized and intracardially perfused with PBS and 4% formalin (Histofix, Roth). WAT was dissected and further fixed for 24 h (obese mice 48 h) in 4 % formalin and treated for paraffin embedding as per standard protocol. Human adipose tissue samples were fixed in 4% formalin for 24 h and embedded in paraffin as per standard protocol. Immunostaining was performed on formalin fixed paraffin embedded sections (3 µm). Sections were deparaffinized, rehydrated, and antigen retrieval was done in Tris-EDTA buffer (pH=9, Cell Signaling 14747). Sections were permeablized in TBS + 0.025 Triton-X for 1 min and blocked in Animal free blocking solution (Cell signaling, 15019) for 1 h. Sections were incubated in primary antibody diluted in Antibody diluent (Cell signaling, 8112) overnight at 4 °C. The following primary antibodies were used: CD31 (1:50, Abcam ab28364); VEGFA (1:100, Abcam ab52917); p53 (1:50, Abcam ab131442); PDGFRβ (1:100, R&D AF1042); Perilipin (1:200, Abcam ab61682); CD31 (1:200, Abcam ab281583); ITM2A (1:50, Abcam ab279387). After washing, sections were incubated with respective secondary antibodies (diluted 1:200 in Antibody diluent) as mentioned in the previous section for 1 h. After washing, sections were counterstained with DAPI for 10 min followed by washing with PBS and mounting in fluorescence mounting medium. Sections were imaged on Axio Scan Slide Scanner Z.1 (Carl Zeiss) or a confocal microscope (LSM 710, Carl Zeiss).

## RNA Isolation from WAT ECs

Sheep anti-rat IgG Dynabeads (Thermo Fisher Scientific) were washed (45 µl/control mice, 90 µl/obese mice) with PBS + 0.1% BSA three times on a magnetic rack. Beads were resuspended in 5 mL PBS + 0.1% BSA and either 45 µL rat anti-mouse CD31 antibody (BD Biosciences, 550274) or 15 µL of rat-anti mouse CD45 antibody (BD Biosciences, 553076) was added (ratio adjusted to the volume of beads). Dynabeads and antibody were incubated overnight at 4 °C on a tube rotator. WAT was excised from mice and minced (without the lymph node). Tissue was digested at 37 °C in a water bath, mixing every 10–15 min (2 mg/mL collagenase II, 2 mg/ml dispase II, 1% FA-BSA in PBS for sWAT; 1 mg/mL collagenase II, 1 mg/mL dispase II, 1% FA-BSA in PBS for vWAT). Homogenates were filtered through a 100 µM cell strainer, washed with PBS + 1% BSA and centrifuged at 300 x *g* for 5 min. Cells were resuspended and again filtered through 40 µm filter and centrifuged at 300 × *g* for 5 min. Cell pellets was resuspended in 3 mL PBS + 1% BSA. CD31-Dynabeads and CD45-Dynabeads were washed three times (5 mL PBS + 1% BSA) on magnetic rack and resuspended in 500 µL PBS + 1% BSA. 500 µL of CD45-Dynabeads was transferred to each cell suspension and tubes were incubated on a rotator (4 °C, 20 min) to negatively select CD45+ cells. The unbound fraction was collected and 500 µL of CD31-Dynabeads was transferred to each cell suspension and tubes were incubated on a rotator (4 °C, 30 min) to positively select for CD31+ cells. Cells bound to Dynabeads were washed three times (PBS + 1% BSA) and once with PBS. After washing, cells were centrifuged (100 x g, 5 min). Cells were immediately lysed and RNA was isolated using the PicoPure RNA isolation kit (Thermo Fisher Scientific).

## cDNA synthesis and qPCR

RNA was measured using a Nanodrop 100 (Thermo Fisher Scientific). A reverse transcriptase kit (Applied Biosystems) was used for cDNA synthesis. cDNA was diluted to 1.25 ng/ul with H2O. mRNA expression was analyzed in duplicates using 5 ng cDNA + SYBR Green Master Mix (Thermo Fisher Scientific) by QuantStudio 3 (Thermo Fisher Scientific). Gene expression was normalized relative to the housekeeping gene *Cph* using the $2^{-\Delta\Delta Ct}$ method. Primers used for qRT-PCR are provided in Supplementary Table 11.

## Scanning electron microscopy

**Tissue.** Mice were anesthetized and intracardially perfused with PBS and fixative (EM fixative as described in ref. 73) as described above.

sWAT was dissected, cut into small pieces 1–2 mm and further fixed in EM fixative for 72 h at 4 °C. Tissues were transferred to 0.1 M sodium cacodylate buffer and stored at 4 °C until further use. For SEM preparation, samples were fixed with 2% osmium tetroxide (OS003, Plano GmbH, Wetzlar) in 0.1 M sodium cacodylate buffer for 2 h at 4 °C and dehydrated by rinsing in 50%, 70%, 80%, 90%, and 100% ethanol. Finally, samples were critically point dried (BAL-TEC, CPD 030). Dried samples were mounted on sample holders and sputter coated (AGB7340m, Agar Scientific). Scanning electron microscopy was performed using a Zeiss Crossbeam 550 (FIB-SEM) operated by Zeiss Smart SEM software (Version 6.07).

**Adipose tissue-derived ECs using MACS.** To generate a single-cell suspension consisting of adipose tissue-derived ECs, WAT was dissected and dissociated as mentioned in the Dynabeads protocol above, until the final washing step before applying the Dynabeads. PEB solution was prepared (47.5 ml MACS rinsing solution and 2.5 ml BSA/EDTA per animal). Next, the supernatant was carefully aspirated, the pellet resuspended with 5 ml PEB and then centrifuged again at $300 \times g$ for 10 min. Meanwhile, LS columns for magnetic separation were equilibrated with 3 ml PEB. After centrifugation of cells, the supernatant was removed, the pellet resuspended in 90 μl PEB and 10 μl of magnetic beads coupled to a CD31 antibody (Miltenyi Biotec; 130-097-418) were added. The Falcon tubes, containing the cell suspension and the magnetically labeled CD31 antibodies, were put onto a rotator in the fridge (4 °C) for 15 min. Afterwards, the cells were washed with 1 ml PEB and centrifuged at $300 \times g$ for 10 min, then the supernatant was taken off and the pellet resuspended in 500 μl PEB. This cell suspension was applied onto a column and washed with 3 ml PEB twice. The columns were removed from the magnetic field and with a plunger, the magnetically labeled cells were washed out with 5 ml PEB onto the second column, to which a MACS SmartStrainer (30 μm) was attached. After the column and the MACS SmartStrainer were washed twice with 3 ml PEB, the magnetically labeled cells were flushed out with 5 ml PEB into a fresh 15 ml Falcon tube, which was centrifuged at $300 \times g$ for 3 min. Next, the supernatant was taken off and the pellet was resuspended in pre-warmed EBM-2 medium, which resulted in 60,000 cells per well, and then incubated at 37 °C and 5% CO2 for 12 h. After 12 h in culture, cells were fixed in 2% glutaraldehyde in 0.1 M sodium cacodylate buffer and stored in 0.1 M sodium cacodylate buffer. For SEM preparation, cells were fixed with 1% osmium tetroxide (Electron microscopy sciences, 19150) in 0.1 M sodium cacodylate buffer for 30 minutes, dehydrated by rinsing the cells in 70%, 80%, 90% and 100% ethanol and finally chemically dried with tetramethylsilane (TMS, Thermo Scientific, 427331000). Samples were mounted on sample holders and sputter coated (Cressington Sputter Coater 108 Auto). Scanning electron microscopy was performed using a Zeiss Crossbeam 550 (FIB-SEM) operated by Zeiss Smart SEM software (Version 6.07).

**Adipokine array**
Adipokine expression in sWAT tissue lysate was detected using the Proteome Profiler Mouse Adipokine Array Kit (R&D, Catalog #ARY013) following the manufacturer's protocol. sWAT lysate was pooled from 4 mice (100 μg lysate from each mice) for each condition, and a total of 400 μg lysate was used on each membrane. Dot blots were imaged on BioRad ChemiDoc MP and analysed using the Image Lab software from BioRad.

**Image analysis**
Image analysis was performed using Fiji software. For vascular density analysis on sections, CD31+ area was measured on whole tissue section and normalized to the area of the tissue to calculate vascular density. For VEGFA expression analysis on sections, VEGFA+ area was measured and normalized to DAPI nuclei count. For PDGFRβ expression analysis,

PDGFRβ expression was measured in areas next to Isolectin B4+ using Enlarge function (set to 5) around blood vessels. PDGFRβ+ cells not in vicinity of blood vessels were not included in the analysis. For ITM2A expression analysis, ITM2A+ area was measured and normalized to CD31+ area.

For whole mount staining, VEGFA+ cells were identified in 3D images using the Spots function in Imaris software. The Surface function in Imaris was used to create surfaces based on PLVAP+ vessels and the distance from PLVAP+ surface to VEGFA+ spots was calculated using the distance transformation. The number of VEGFA+ cells that were less than 10 μm or more than 10 μm away from the PLVAP surface was analyzed. Volume of PLVAP+ vessels was calculated from the surface based construction and normalized to Isolectin B4 volume.

**Statistical Analysis**
GraphPad Prism v10.4.2 was used to generate graphs and for statistical analysis. Groups were tested for normality. Statistical significance was calculated for two unmatched groups with normal distribution by unpaired two-tailed Student's t-test with Welch's correction (adjusted for multiple comparisons). Mann-Whitney test was used for data, which did not pass normality testing. Data sets are presented as mean ± SEM. For panels 6e, j, Stata 18 (StataCorp, College Station, TX, USA) was used. Significance was established at $P < 0.05$ by linear regression with cluster-robust standard errors.

**Reporting summary**
Further information on research design is available in the Nature Portfolio Reporting Summary linked to this article.

## Data availability
The scRNA-Seq data generated in this study along with processed files, have been deposited in BioStudies database under accession code E-MTAB-12855. All data supporting the findings described in this manuscript are available in the article, in the Supplementary Information, and from the corresponding author upon request. Source data are provided with this paper.

## Code availability
The code used for bioinformatics analysis can be accessed at https://doi.org/10.5281/zenodo.15402419[74].

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

## Acknowledgements

We thank Genentech for the *Vegfa* floxed mice and Tiago V. Pereira for advice on statistical analysis on data for Fig. 6. We are grateful for support by the Single-cell Open Lab, Light Microscopy, FACS, and High-throughput sequencing core facilities and the animal facilities in DKFZ, University Medical Center Göttingen, and at the UCL Institute of Ophthalmology. We thank S. Köhler (Center for Advanced Imaging, CAI, Heinrich Heine University Düsseldorf) and A. K. Bergmann (Core Facility, UKD Düsseldorf) for their support with SEM. This work was funded by the Deutsche Forschungsgemeinschaft (DFG) project no. 394046768-SFB1366 (A.F., S.D., and R.H.A.) and the British Heart Foundation project no. PG.23.11342 (M.R., C.R.).

## Author contributions

Conceptualization: S.S.H., C.R., A.F., Methodology: S.S.H., D.J., M.R., I.A., D.E., I.M., J.T., C.K., Mv.H., L.C.C., R.H.A., E.L., J.K., C.R., S.D., A.F., Investigation: S.S.H., D.J., M.R., I.A., D.E., I.M., J.T., C.R., Visualization: S.S.H., Funding acquisition: A.F., M.R., C.R., Project administration: A.F., Supervision: A.F., Writing–original draft: S.S.H., A.F.

## Funding

## Competing interests

C.K.—Roche: Employment, patents/royalties, stock ownership. The remaining authors declare no competing interest.
