## [Transparent Peer Review file · Nature Communications]

Obesity drives depot-specific vascular remodeling in male white adipose tissue

Corresponding Author: Professor Andreas Fischer

Version 0:

Reviewer comments:

Reviewer #1

(Remarks to the Author)

The authors have addressed all my comments and concerns.
The manuscript has improved significantly since the original submission.

Reviewer #2

(Remarks to the Author)

No further comments.

Reviewer #3

(Remarks to the Author)

We thank the authors for the thorough revisions made to the manuscript. In this revised format, the manuscript meets the requirements for a “research article” with respect to clarity in data analysis, data deposition, and code availability. The datasets previously noted as missing for a “resource article” have been verified as deposited in the repositories specified by the authors, and an extended analysis code has also been provided. The clarity of the manuscript has also notably improved, particularly in the way specific comparisons are described across different sections. The authors have also addressed all the points raised in the 'Additional Questions' section.

The only remaining point to highlight concerns the biological significance of the findings. In response to the related comment, the authors acknowledge that “it could very well be that they [fenestrated endothelial cells in adipose tissue] do not have a role in systemic endocrine or metabolic homeostasis but a more local effect in interacting with stromal cells in the tissue.” The authors also note the current technical limitations that make specific functional studies of this subpopulation not possible due to the lack of suitable genetic tools. This is a fair point and reflects the current state of the field. Nevertheless, it does mean that the broader physiological role of these cells remains uncertain at this stage. That said, the identification of these cells in human tissue provides added value to the study. These aspects contribute to the manuscript’s overall merit, even if some biological questions remain open for future investigation.

Reviewers' comments:

Reviewer #1 (Remarks to the Author):

Only the comments that require further follow-up are mentioned in the response to previous comments. This is followed by new comments.

Author responses start with >>

RESPONSES TO PREVIOUS COMMENTS

PREVIOUS COMMENT 1) Importance and the technical details of the use of cell-hashing in single-cell RNA sequencing and the analysis pipelines:

Single-cell RNA sequencing has been widely used in the recent years for example to understand cellular heterogeneity and transcriptomic changes in disease. One main caveat of this method is the false positive differential expression values it may produce in high-expression genes, when comparisons are made between just two samples. To avoid this, increasing the number of samples in each condition to achieve more robust statistics can be employed. In this manuscript, the authors use a more recent technique called cell hashing (or multiplexing), which allows more robust statistical analyses and overcoming high-expression gene bias. However, the importance and the advantages of this method are not clearly stated throughout the manuscript. Moreover, a very important aspect, the technical pipeline of how the results were achieved is not described in sufficient detail, as the order of steps in the analysis pipeline may yield different results. Finally, the raw data was not accessible to the reviewers to assess the findings. The authors should clarify how the analyses were performed, which steps were followed in which order, how the statistics were performed, how the data were integrated, and make sure the data are available to reproduce the results.

>>We thank the reviewer for this comment. The methods part for the bioinformatics section was updated and now we describe the bioinformatics pipeline step-by-step. We have uploaded all the source code generated throughout the study to a GitHub repository. This is available under: <https://github.com/djhn75/Cite-Seq-Heidelberg>.

>>Additionally, we have uploaded all the datasets to ArrayExpress under the accession E-MTAB- 12855. All raw-data files along with the processed expression matrices with all metadata information's can be found in an R-object.

REVIEWER RESPONSE 1) Thank you for clarifying that the datasets have been uploaded to ArrayExpress. While the provided Seurat object may suffice for verifying some of the findings by the reviewers, it does not fulfill the standard requirements for raw data in a resource paper. Raw data typically refers to unprocessed files such as FASTQ files, which are necessary for

fully reproducing the analysis pipeline from the start. These files should have been uploaded to the repository to ensure transparency and reproducibility.

Moreover, the Seurat object lacks essential details, such as raw cell classifications and metadata for certain analyses (information on cell cluster labeling, as one example), making it challenging to verify all findings described in the manuscript. For a resource paper, the raw data should be made publicly available in universally accessible formats, accompanied by sufficient instructions for use.

We apologize for this inconvenience. The processed Seurat object has been updated and now contains all the requested information, like the cell types and other information mentioned in the paper. Please use this link to get access to all relevant information:

<https://www.ebi.ac.uk/biostudies/arrayexpress/studies/E-MTAB-12855>

Additionally, the source code for the data analysis has been updated and made easier to understand:

https://github.com/djhn75/Cite-Seq-Heidelberg/blob/main/2.Cite-SEQ_Analysis.Rmd

All the information regarding clustering, doublet detection and cell type annotation are included in the Github repository. Additionally, all the preprocessing commands were added to the Github Repository

<https://github.com/djhn75/Cite-Seq-Heidelberg/blob/main/1.preprocessing.sh>

The authors should confirm that the raw FASTQ files are available and clarify where and how they can be accessed. Without this information, the manuscript does not meet the community standards for resource publications.

All the Raw Data FASTQ files together with an R-Object file, containing the processed and integrated samples are uploaded at BioStudies under the accession number (**E-MTAB-12855**). We additionally uploaded the raw gene/cell counts and the cite-seq counts to this repository.

Please see also section 6 where we detail our attempt to reproduce the analyses and what kind of information was missing.

We highly appreciate this. All of our answers are outlined in section 6.

PREVIOUS COMMENT 2) Overall clarifications of comparisons:

In the results section, the authors describe differential gene expression/changes they find. Since there are two tissues, two conditions, and several clusters they study, it is essential to clarify the comparisons in each paragraph and also in the figure legends. The authors sometimes use cluster numbers and later denoted names for the clusters. The clarifications should be done for example as in lines 176-177.

>>We apologise for the inconvenience. We have now included text in the manuscript as well as in figure legends to describe which cell types are being compared under different conditions. We have also replaced all cluster numbers with cell annotations.

REVIEWER RESPONSE 2) Thank you for the revisions and for attempting to clarify the comparisons in the text and figure legends. However, several aspects remain unclear, particularly concerning the comparisons and the interpretation of differential gene expression (DEG) analyses.

As an example, on line 160, the authors state that they focus on "obesity-driven changes in different capillary beds" but also mention comparing "one cluster against all other clusters." This creates confusion about whether they are referring to cluster markers (genes that define a cluster) or DEGs between conditions. The current wording lacks clarity, making it difficult to follow the rationale behind the analyses.

For example, in the case of Capillary EC I and II, it appears that the authors infer from cluster abundances that Capillary EC II is more prevalent in obesity and then use cluster markers to define DEGs. If this is the intended approach, the methods and logic for defining DEGs require explicit clarification in the manuscript. Are the authors comparing gene expression within a specific cluster across conditions, or are they comparing markers of one cluster to another? This distinction is critical for accurately interpreting the findings.

Furthermore, despite the added clarifications, there are still ambiguities in the results section and figure legends regarding exactly what is being compared. The manuscript would benefit from a clear and consistent description of (1) the cell types or clusters being compared, (2) the conditions under which these comparisons are made (e.g., obesity vs. control), and (3) whether the analysis pertains to cluster markers or DEGs across conditions. Revising the text and legends to address these points explicitly would greatly improve the readability and scientific rigor of the study.

We had already replaced cluster numbers with annotated cell types in the previous version of the manuscript as well as specified which conditions and tissue are being compared; obese or lean; sWAT or vWAT. As per the reviewer's suggestion, we have also clarified whether we used cluster markers or DEGs for analysis.

PREVIOUS COMMENT 4) The biological significance of PLVAP+ ECs in sWAT:

Significance of PLVAP+ cells in sWAT: The biological significance and function of these cells in sWAT is not clarified in this study. The authors only speculate about the significance/function and the consequences of their disappearance (lines 386-390) of these cells in sWAT. The authors mention an earlier study that also identified the fenestrated endothelial cells in sWAT and showed their reduction in obesity. If these cells are essential in an endocrine function as speculated, the authors could for example target VEGFA and see if the fenestrated ECs in sWAT indeed disappear. And if the cells disappear, would the levels of sWAT produced "endocrine" factors decrease in serum? The authors show that these cells are found in human sWAT, are their numbers reduced in human obesity? There are several other studies investigating adipose tissues using single cell and single nucleus RNA sequencing in humans with and without obesity. The authors can re-analyse these publicly available data and discuss further the correlation with human disease.

>>To address the first point regarding targeting VEGFA and see if the fenestrated ECs in sWAT disappear, we performed systemic VEGFA blockade using a well-established VEGFA blocking antibody (B20.4.1 from Roche). Since, VEGFA blockade can cause secondary phenotypes, we did a short-term treatment (two weeks) and found reduction in PLVAP+ ECs

(Fig 5m-q, see figure below). We also complemented this study using genetic Vegfa loss-of-function experiment and saw similar results (Fig 6, see figure below), while in Vegfa gain-of-function mice there were more PLVAP+ ECs (Fig 6, see figure below) Regarding whether the levels of sWAT produced endocrine factors reduce in serum in above experiments, we could not perform any conclusive experiments as the adipokine milieu secreted by sWAT and vWAT are similar. Hence, a serum analysis would not provide conclusive results on whether a particular adipokine was secreted by sWAT or vWAT. In addition, VEGFA signalling is important for various aspects of adipose tissue biology. Hence, blocking it could also affect production of these adipokines in adipocytes. Therefore, again, we cannot conclude on whether lower levels of a particular adipokine in serum are due to less production or impaired transport into the blood stream.

>>Concerning this cell population in human adipose tissue, we analysed a recently published integrated atlas of human subcutaneous white adipose tissue (<https://www.biorxiv.org/content/10.1101/2024.09.22.610444v1>). We detected expression of PLVAP and ITM2A (another top marker gene in the fenestrated EC cluster in our murine dataset) in two distinct clusters (see figure below, UMAPs not included in manuscript). Upon immunohistochemical staining of ITM2A, from lean and obese human male donors, we observed a reduction in ITM2A expression in obese human sWAT (Fig 6k, l, see figure below).

REVIEWER RESPONSE 4) The authors performed additional experiments involving systemic VEGFA blockade and genetic VEGFA loss-of-function, demonstrating a reduction in PLVAP+ endothelial cells in sWAT. However, the systemic nature of these approaches -as was mentioned in the original reviewer comment- introduces confounding variables, as VEGFA signaling is crucial for vascular beds rich in fenestrated endothelia, such as those in the small intestine, liver, and pancreatic islets, which might affect weight gain and metabolism. Notably, previous studies have shown that the main effector VEGFA receptor 2 (VEGFR2) deletion in endothelial cells can significantly impact these vascular beds within 48 hours of tamoxifen induced gene deletion (PMID: 35050301).

We are very well aware of the systemic effects of VEGFA blockade on different vascular beds. Our groups are working in the field of vascular biology for more than 20 years and several key findings regarding angiogenesis, angiocrine signaling and tumor angiogenesis were made by us in the past.

Keeping that in mind, and after review of literature we had designed our experiment with lower dosage and for shorter period. The VEGFR2 deletion study that the reviewer is referring to (PMID: 35050301), where they see affects within 48 hrs was performed on triple VEGFR knockouts where they had deleted all the VEGFR1/R2/R3 in a single adult mice. In the same study in single VEGFR2 deleted animals, there are affects after 8 days of tamoxifen induction, but we have to consider that these are homozygous floxed animals with complete loss of VEGFR2 in the endothelium and therefore the affect is more dramatic. While in our VEGF antibody blocking experiment, we limit the dosage and in our Vegfa genetic loss of function, we used **heterozygous** Vegfa floxed mice to mitigate these systemic affects. Nonetheless, there is no arguing that there will be some non-specific affects to Vegfa deletion/blockade as it is a master regulator of multiple biological processes but we have tried to address this in the best possible manner. Currently, in the vascular field there are only few organ-specific promoters to induce deletions in specific organ. Almost every study in the field relies on the use of Cdh5 driven Cre lines (and also Tie 2 driven Cre lines) which systemically delete the gene of interest from nearly all vascular beds and this would definitely have some systemic

effect. This is state-of-art and currently the accepted way of conducting experiments until more specific genetic targeting becomes available.

Systemic VEGFA inhibition has also been shown to reduce vWAT vascular density in studies utilizing VEGF-Trap or Ad-sVEGFR1 for 10–14 days (PMID: 16172168). These prior findings emphasize the broad and immediate effects of VEGFA inhibition, leaving the interpretation of adipose-specific outcomes unclear.

Regarding the second study which the reviewer is referring to (PMID: 16172168), the authors demonstrate loss of fenestrations in different organ beds upon VEGF blockade. We would like to clarify that they do not show loss of fenestrations in epididymal fat (equivalent to visceral fat in our case) or subcutaneous fat by electron microscopy upon VEGF signaling blockade. They do see reduction in vascular density in epididymal fat in a sporadic manner and we would like to directly quote from the paper.

“Capillary loss was uniformly distributed and resulted in a generalized thinning of the microvasculature in all affected organs with the exception of epididymal adipose tissue, in which capillary regression was patchy, ranging from 51% to none, with an average decrease of 25%.”

“A small and regionally variable dropout of capillaries was found in adipose tissue. Although some capillaries in fat are fenestrated, they are not the dominant phenotype, and the number of fenestrations appears to be relatively small. Therefore, we cannot exclude that some VEGF-dependent normal capillaries lack endothelial fenestrations.”

While we understand the reviewer’s concern regarding broad effects of VEGF signaling blockade and we hope we could provide some clarifications regarding differences between the studies cited and our experiments.

While the authors demonstrate the sensitivity of PLVAP+ ECs in sWAT to VEGFA signaling, this observation is not novel. The presence of fenestrated capillaries in subcutaneous adipose tissue has been previously described (PMID: 11344271), so as the sensitivity of fenestrated capillaries to the lack of VEGF signaling (PMID: 16172168). A critical gap remains in clarifying the biological significance of these cells, particularly in the context of obesity. For instance, what functional role do fenestrated ECs in sWAT play in endocrine or metabolic homeostasis, and how does their loss impact obesity progression or adipose tissue function?

Yes, the presence of fenestrated capillaries in subcutaneous adipose tissue has been previously described in this single study 20 years ago (PMID: 11344271), using transmission electron microscopy. However, the use of current single-cell methodologies allowed us to understand the biological signature of these cell types, understand and predict putative signaling pathways involved in maintenance of fenestration in subcutaneous adipose tissue. We would again like to recapitulate that even though Bondareva et al (PMID: 36400935) identified fenestrated ECs in subcutaneous adipose tissue, they did not describe or perform any type of experiments to validate or characterize this cell population. While we have worked extensively on this cell population utilizing various tools as already described. Furthermore, the identification of this fenestrated signature allowed us to stain for this population in human white adipose tissue.

Regarding the sensitivity of fenestrated endothelium to lack of VEGF signaling (PMID: 16172168), we have already addressed this in the previous section that this study does not show any link between fenestrations and subcutaneous adipose tissue. The study describes effect of VEGF inhibition on fenestrated capillaries in other vascular beds.

Regarding the functional role of these cells types in endocrine or metabolic homeostasis, we would again like to point out that the proportion of this population is roughly 15% of all endothelial cells in subcutaneous adipose tissue. It could very well be that they do not have a role in systemic endocrine or metabolic homeostasis but a more local affect in interacting with the stromal cells in the tissue. Technically, it is simply impossible at the moment to specifically target this subpopulation using the whole battery of genetic mouse models which are present worldwide.

The authors hypothesize an endocrine function for these cells but provide no experimental evidence to support this claim. While they note that VEGFA blockade could influence adipokine secretion or transport, no targeted experiments were conducted to verify whether fenestrated EC loss alters sWAT-derived adipokine levels or systemic metabolic outcomes. These targeted experiments are understandably difficult, as they would require cell-type-specific manipulations or more sophisticated models to disentangle the systemic effects of VEGFA signaling from adipose-specific phenomena. However, without such evidence, the functional importance of fenestrated ECs in sWAT remains speculative.

We understand that not being able to establish the functional role of fenestrated endothelial cells is a caveat of our study and we have addressed putative functions of this population in the Discussion section of our manuscript. At this moment, there are no technically sound experiments that could yield a convincing result. As the reviewer mentioned that it would be difficult to address this question *in vivo* (due to lack of specific promoters) and there are significant caveats in performing experiments related to fenestrations *in vitro*. Fenestrations are very dynamic in nature and not easily maintained (PMID: 39201792). Also, unlike other specialized organs, where all endothelial cells exhibit fenestration, subcutaneous adipose tissue maybe has approx. 15% endothelial cells as fenestrated, which might be even further reduced *in vitro*, as fenestrae are lost over a period of time in culture, making it difficult to do any functional assay *in vitro*.

Regarding not performing any targeted experiments to check for sWAT-derived adipokine or systemic metabolic outcomes, we already have plasma samples collected from all our mice experiments and would be happy to check for a sWAT-derived adipokine if the reviewer can suggest one. Our review of literature has not yielded any good candidates that could be tested. While there are differences in levels of adipokine secreted from subcutaneous vs visceral fat but so far nothing, that is exclusive to one of the depots.

Thus, while the study confirms previously known phenomena, it does not advance understanding of the biological role or consequence of fenestrated EC loss in sWAT. This omission represents a significant limitation, as the novelty of this work would lie in elucidating the specific relevance of these cells in adipose tissue biology, particularly under obese conditions.

While we acknowledge the lack of functional significance in our study, there are several novel aspects to the study as described below.

1. **Conceptual novelty of the study:** We understand that our manuscript is foreshadowed by Bondareva et al. (PMID: 36400935) who performed a single-cell transcriptomics experiment to understand the impact of obesity on endothelial cells obtained from several organs including subcutaneous and visceral white adipose tissue (WAT). We would like to take this opportunity to address again why our manuscript, despite using a similar approach to the transcriptomics screen offers a much more detailed and novel insights into WAT endothelial cell biology in obesity.
 - i) The WAT part was addressed very briefly (half a main figure) in their manuscript without going into details about each cell type, activated pathways or any upstream regulators while we have addressed each of the annotated cell type with respect to above mentioned aspects in detail in our manuscript. Fenestrated endothelial cells are annotated and identified in their figure but there is no mention of it at all in the main text in their manuscript while we have used several modalities like **whole mount imaging, validation via reporter mice** as well as **ultrastructural analysis** via electron microscopy to establish the existence of this cell type. Also, our single-cell screen included **Cell Hashing**, which has not been used with endothelial cells before.
 - ii) In addition, we identified **VEGF signaling pathway**, source of VEGFA as well as performed several **loss and gain of function experiments** to understand the impact of VEGF signaling on fenestrated ECs in adipose tissue. We have included **multiple time-points** in our study to understand the progression of vascular rarefaction during obesity as well as the **VEGF response at different time points**. Apart from changes in vascular density, we have further characterized the vasculature in obese WAT in terms of **mural cell coverage and p53 activation**.
 - iii) Lastly, we have also included **data from human subjects** in our study to understand whether the fenestrated endothelial cell population has any translatable potential.

In conclusion, the parallels between our study and the Bondareva study stop at the bioinformatics screen. All the above-mentioned points highlight the novel elements in our manuscript, and we hope that we are able to convince you about the conceptual novelty and the significance of this study as an in-depth resource of how obesity impacts WAT endothelial cells. As suggested by one of the reviewer we have now added more text to highlight these points in the manuscript.

NEW COMMENTS:

5) Discrepancy in Depot-Specific Expression of PLVAP

The authors suggest that PLVAP-expressing fenestrated endothelial cells represent a sWAT-specific cluster affected in obesity. However, Plvap is reported as a top 10 cluster marker in venous ECs, with expression in over 95% of these cells (Supplementary Table 6). This is

further supported by the feature heatmap (Extended Data 5d), which shows Plvap expression in both sWAT and vWAT.

Our own data show clear PLVAP expression in gonadal WAT of female mice (Figure 1 in attached document), indicating that fenestrated endothelial cells are not restricted to the subcutaneous depot and may also be present in other depots. Could the authors clarify this apparent discrepancy? Is it possible that the lack of detection in vWAT is due to staining issues or other methodological factors? Further explanation would strengthen the conclusions about depot specificity.

We are aware of Plvap expression in venous ECs in our dataset. However, expression of Plvap does not translate to presence of fenestration. Plvap is a component of diaphragm protein and can be present in caveolae, fenestrae and transendothelial channels (PMID: 36781482). Therefore, we had performed an ultrastructural analysis to specifically look for fenestrae in endothelial cells. We annotated this cell type as fenestrated not just due to the expression of Plvap but also based on congruencies in gene signature from other published dataset (PMID: 32059779) for fenestrated endothelium (in choroid plexus) including gene expression of Esm1, higher Nrp1, etc, which we also validated using Esm1-reporter mice.

In subcutaneous WAT, when we stain for PLVAP we do observe some larger caliber vessels which are PLVAP+ (could be PLVAP+ veins), but most of the staining is in smaller caliber vessels. Since whole mount staining is not very throughput, as we can only image a very small part of the tissue, we had come up with a strategy of using a combination of Podocalyxin and Claudin5 staining to locate fenestrated endothelium in big tissue sections. However, the reviewer themselves suggested in their last comments that we use PLVAP exclusively for staining and not Podocalyxin and Claudin5. We do encounter occasional large caliber vessels that could be veins but since they are present at a similar frequency in control and obese mice, this does not affect the overall analysis.

We only detected PLVAP+ vessel (large caliber) in one image out of 37 images that we had acquired for gonadal fat probably due to their lower proportion in gonadal vWAT. It could be that the Plvap+ cells in the reviewer's dataset are venous endothelial cells or there could be sex-specific differences, since we have worked exclusively with male mice and the reviewer's data set is from female mice.

6) Attempt to Reproduce Data Analysis - missing information

We attempted to re-analyze the provided Seurat object downloaded through the provided accession number. While we tried our best to follow the code from the authors there were missing details in the authors' methods, which made reproducing their analysis challenging. Below is a detailed breakdown of our steps, observations, and questions:

We apologize that the reviewer had problems validating the data. We updated and restructured the submitted source code file to simplify the validation of the cluster. For the sake of completeness, we have uploaded all cells in the previous submission. So, the uploaded processed data file contained doublets and unmapped antibody reads.

However, if we download the RDS file, subset the singlets, reperform the clustering and then remove clusters with >50% doublets, we get exactly 12 clusters as stated in the paper.

Code to recluster the published dataset:

```

library(ggplot2)
options(timeout=10000)
download.file("https://www.ebi.ac.uk/biostudies/files/E-MTAB-
12855/AllSample.SeuratObject.AllCells.Rds", "MTAB12855.RDS")
MTAB12855<-readRDS("MTAB12855.RDS")

MTAB12855.single<-subset(MTAB12855, subset =
HTO_classification.global=="Singlet")
DefaultAssay(MTAB12855.single)<-"RNA"
MTAB12855.single <- NormalizeData(MTAB12855.single)
MTAB12855.single <- FindVariableFeatures(MTAB12855.single, selection.method =
"mean.var.plot")
MTAB12855.single <- ScaleData(MTAB12855.single, features =
VariableFeatures(MTAB12855.single))
MTAB12855.single <- RunPCA(MTAB12855.single, features =
VariableFeatures(MTAB12855.single))

#detect doublets
nExp_poi <- round(0.075*nrow(MTAB12855.single@meta.data))
MTAB12855.single<-doubletFinder(MTAB12855.single, PCs = 1:10, pN = 0.25, pK =
0.09, nExp = nExp_poi, reuse.pANN = FALSE, sct = FALSE)
MTAB12855.single$DF.classifications<-
MTAB12855.single$DF.classifications_0.25_0.09_1113
MTAB12855.single.noDoublets<-subset(MTAB12855.single, subset =
DF.classifications=="Singlet")

#recluster after doublet removal
MTAB12855.single.noDoublets <- RunPCA(MTAB12855.single.noDoublets, features
= VariableFeatures(MTAB12855.single))
MTAB12855.single.noDoublets <- FindNeighbors(MTAB12855.single.noDoublets,
reduction = "pca", dims = 1:10)
MTAB12855.single.noDoublets <- FindClusters(MTAB12855.single.noDoublets,
resolution = 0.6, verbose = FALSE)
MTAB12855.single.noDoublets <- RunUMAP(MTAB12855.single.noDoublets,
reduction = "pca", dims = 1:10)

table(MTAB12855.single.noDoublets$seurat_clusters)
table(MTAB12855.single$seurat_clusters)
df<-as.data.frame(as.matrix(table(MTAB12855.single.noDoublets$seurat_clusters,
MTAB12855.single.noDoublets$DF.classifications)))

ggplot(data=df, aes(x=Var1, y=Freq, fill=Var2)) + geom_bar(stat="identity")

```

Resulting clusters:

We assume that reviewer 1 did not subset the data correctly, because if we do not subset the data we also get 17 clusters. Without having exact source code, it is not possible for us to reproduce the results from the reviewer.

Data Preparation

- > Downloaded the provided Seurat object, which appears to be pre-integrated.
- > Subsetted the data to include only cells labeled as "Singlet" based on the metadata column HTO_classification.global.
- > Resulting dataset contained 14,843 cells.

Doublet Detection Using DoubletFinder

Normalization and Scaling: The methods section did not describe the exact steps for running DoubletFinder. As such, I normalized and scaled the data before running the analysis, following standard practices.

Parameter Optimization:

- > Used DoubletFinder's default protocol (no ground-truth data) to identify the optimal pK value, which was determined to be 0.3 based on the highest BCmetric value (see Plot 1 in attached document).
- > The value for detecting homotypic doublets was not specified in the methods. I assumed a 60% threshold, considering the dataset represents a homogenous cell population (endothelial cells). Homotypic doublets are typically more frequent in such cases.
- > Set the expected doublet rate to 12%, as 10X Genomics experiments generally yield ~10% doublets per 10,000 cells.
- > Set pN to 0.25, as per the DoubletFinder guidelines.

Results:

- > Predicted doublets were primarily concentrated in clusters 9 and 14 (see Plot 2 in attached document).
- > DoubletFinder-predicted doublets were excluded, resulting in 14,131 cells (712 cells removed).

Clustering and Feature Selection

>Calculated the most variable features using the mean.var.plot method, identifying 2000 features.

>Re-clustered the data using:

o 10 principal components (dimensions).

o Resolution of 0.6.

Outcome: The reclustering yielded 17 clusters in total.

Questions and Missing Information

1) DoubletFinder Parameters:

-What specific parameters were used in the authors' DoubletFinder analysis?

We ran doublet finder with the following parameters:

```
nExp_poi <- round(0.075*nrow(NUMBER_OF_CELLS_IN_SAMPLE))  
PCs = 1:10, pN = 0.25, pK = 0.09, nExp = nExp_poi, reuse.pANN = FALSE, sct = FALSE
```

So we assumed 7,5% doublet formation rate and also kept all the other parameter as stated in the doubletfinder guidelines (<https://github.com/chris-mcginnis-ucsf/DoubletFinder>)

-How was the homotypic doublet proportion determined?

Homotypic doublet proportion was not used in this dataset, as we determined clusters on individual sample level and performed the annotation afterwards on the integrated dataset. So we did not have a valid annotation at the time of Doublet detection.

2) Clustering Resolution:

-What resolution and principal component dimensions were used for clustering?

-What criteria were applied to define cluster markers and exclude doublet clusters?

3) Integration and Preprocessing Details:

-The data was already integrated upon download. Could the authors provide details on the integration method used?

-Was additional filtering or preprocessing performed before downstream analyses?

All these requested details are clearly stated in the method section in the paper and can also be found in the submitted source code. Yet, the resolution for clustering was set to 0.6 and the first 10 PC's were used for clustering. Besides doublet removal no additional filters were applied and no clusters were excluded from the analysis.

4) Discrepancy in Clusters:

-Our re-analysis yielded 17 clusters, while the manuscript reports a different number. Could the authors clarify their approach to defining clusters and the criteria for cluster exclusion or merging?

We tried hard to recreate the results from the reviewer, however if we download the published dataset, filter and recluster, we get 12 clusters as stated in the paper. The R code for our reanalysis can be found in the answer of Point 6. We assume that the reviewer can confirm our results when running the above source code.

Summary: While we were able to analyze the dataset to some extent, the lack of detailed descriptions in the methods section prevented us from reproducing all aspects of the authors' pipeline. Clarification of these details would greatly improve the reproducibility of the study, considering this is a resource paper.

We partially agree with the reviewer that the submitted datasets were missing some important information like the assigned cell types and that the source code was not straight forward. We therefore updated the processed R-Data file, so it only contains the final cells and the assigned cell types. Additionally, we updated the source code repository so that is easier to recreate the results from our paper and we uploaded the matrix files after Starsolo and cite-seq Count.

However, we disagree with the statement that a reanalysis of our published dataset results in 17 clusters. In our hands we could not recreate those results.

Additional comments & questions

- In general, it has been challenging to identify the specific changes made by the authors during the revision process, as there were no page or line indications provided as to where the changes were added and corrections were made, nor was a "tracked changes" document included. Providing such a document would significantly aid reviewers in efficiently assessing the revisions and ensuring all concerns have been adequately addressed.

This was not required by the Nature journal group and therefore we did not generate one. Since we did so many changes in the text, track changes would not really help to improve legibility.

- In the introduction, the authors state that they study endothelial cells (ECs) at single-cell resolution "over the course of obesity." However, they report only results from the 8-week time point (line 76). Could the authors clarify this discrepancy?

We have removed the word "course".

- What is the context or relevance of p53 levels to the rest of the results? This connection is unclear in the text (line 92).

This was a requirement by another reviewer. p53 upregulation is linked to cellular stress and senescence and was shown to be upregulated in endothelial cells in some vascular beds in response to high calorie diet (PMID: 24857662). We have now added some text to explain the connection.

- For the scRNA-seq resource provided, it would strengthen the manuscript to include comparisons with previously published, similar datasets. For instance, do the same clusters and cluster markers emerge, and how do they align with the authors' findings?

We already mentioned the Bondareva et al dataset before. Now we have added a short paragraph comparing our results with their's In Discussion

- The comparison of venous and arterial EC abundances under different conditions is based on scRNA-seq data with a relatively low number of cells (line 153). Could the authors comment on how this limitation might impact their conclusions?

We understand the reviewers concern regarding low number of cells for these subtypes but since we used cell hashing we could add some statistics (n=3 mice in each condition) and this data is significant (for venous ECs). Furthermore, we have not really made any significant conclusions for this piece of data nor pursued this in detail in our manuscript.

- Both Capillary I and II populations are present in lean and obese tissue scRNA-seq data, yet the authors claim that Capillary I represents the homeostatic state of ECs in lean adult mice (line 163). Could the authors provide more evidence or rationale for this conclusion?

Yes, they are present in both conditions. However, Cap I is the dominant subtype in lean vWAT (more than 40%) in comparison to Cap II, which is less than 15%. This was our rationale behind this statement. However, we have now removed this statement from the manuscript and also adjusted the graphical abstract accordingly.

- The term “most notable processes” (line 178) is ambiguous. Does it refer to the most significant, most abundant, or most interesting processes? The basis for this designation should be clarified.

It is clear from the figure that these are some of the most upregulated processes. However, if the reviewer found it ambiguous, we have now changed it to “most significant”,

- Why did the authors analyze only upregulated genes in their differential gene expression (DGE) analysis? Including downregulated genes could provide a more comprehensive understanding of the biological changes.

We thank the reviewer for their comment and agree that including the downregulated genes would provide a more comprehensive understanding. Pathway and GO term analysis for some groups did not show much depot specific or EC subtype specific differences (see below).

sWAT Arterial ECs downregulated pathway

sWAT Venous ECs downregulated pathway

vWAT Arterial ECs downregulated pathway

vWAT Venous ECs downregulated pathway

However, we did see some interesting results for Cap II population in vWAT. We have now included some of these data in Extended Figure 6 (see below).

- The elevated expression of Hif1a is reported in the HFD scRNA-seq samples. However, Hif1a is an acute response gene, raising the question of whether its expression would persist at 8 weeks of HFD exposure (line 189). Could the authors address this potential inconsistency?

It is correct that Hif1a is an acute response gene but we have not implied anywhere in the text that its expression “persists” at 8 weeks of HFD, which would suggest an earlier upregulation and maintenance until 8 weeks of HFD. We just observe a predicted upregulation at 8 weeks

of HFD. It should be noted that it could take some time between the start of the diet and the hypoxia to set in, followed by compensatory angiogenesis. However, at a later period of time the rate of angiogenesis cannot keep up with rapidly expanding adipocytes resulting in hypoxia. Therefore, it is not surprising that we had this observation at 8 weeks of HFD.

Reviewer #2 (Remarks to the Author):

The authors' effort to consider most of our major and minor comments are appreciated. Although the experiments and rationale of the paper in its current state are acceptable, one final modification would increase readiness for publication in this journal.

Major comment 1 needs to be adequately addressed. While the Bondareva reference was added, only a sentence acknowledging it was included in the paper. The authors should name the similarities found (e.g., HIF signaling pathways and ECM remodeling) and the differences they mentioned to answer my previous comments. Mention that some of the paper's findings are aligned with already peer-reviewed work, and not only the ones in the BioRxiv (which are not peer-reviewed) will substantially support the paper's conclusions. Moreover, it would be greatly appreciated if the authors include what is already known in the literature in other contexts about the relationship of Plvap and VEGFA, and stress the differences between this manuscript and already published work. Overall, the authors should describe more sharper / more completely what is not novel in the paper, refer to the literature, and emphasize the novelty of the manuscript.

We thank the reviewer for their positive evaluation of our work. We have included the suggested text modification in the new version of our manuscript. In addition to relationship between VEGFA and PLVAP, we have now also added a more detailed comparison with the Bondareva paper in the discussion section while also highlighting the novel aspects of our study.

Reviewer #3 (Remarks to the Author):

In this manuscript, Hasan and colleagues explore the impact of obesity on ECs in the sWAT and vWAT. Their revised manuscript has improved significantly. While not without shortcomings, the experiments with VEGFA provide evidence that VEGFA is likely to be important for maintaining fenestrations in subset of sWAT ECs. Similarly, reduction of fenestrated ECs in human sWAT in obesity strengthens the authors' conclusions. This has added much needed novelty to the manuscript.

My comments and concerns below pertain to the new data that was added during the revisions:

1. The new human data showing reduced fenestrated ECs in obesity in sWAT strengthens part of the authors' conclusions. A similar staining (or analysis of human scRNA-seq data) on vWAT should be provided to show whether fenestrated PLVAP+ ECs are also specific to sWAT in human.

We thank the reviewer for their comment. We have now performed ITM2A stainings on human vWAT. We do detect ITM2A+ ECs in human vWAT. This population is also reduced in vWAT from obese human subjects. We have now included this data in Figure 6m, n (see figure below). It is important to understand that several differences exist between mice and human WAT in terms of location, vascularization, adipocyte size and immune cell milieu (PMID: 38086922). Visceral WAT in mice is mostly perigonadal whereas the most accessible and studied visceral WAT in human is omental WAT. Human vWAT is more vascularized with smaller adipocytes compared to human sWAT (PMID: 27818937), while the opposite is true for mice. Therefore, given the existing biological differences it is not surprising that we observe different patterns in terms of fenestrated ECs between mice and human. We have added these points in the Discussion section also.

2. The authors provide new EM data showing the presence of fenestrations in FACS isolated sWAT ECs (Fig. 4d). However, it is not clear how many cells or animals were assessed (no replicate information provided)? Were replicates undertaken? How many ECs were analyzed? What proportion of ECs displayed fenestrations? The vWAT ECs should be provided as a control. Do they show any fenestrations via EM? These analyses are critical for understanding the significance of the new data.

We first attempted to do scanning electron microscopy on whole white adipose tissue. However, presence of extensive ECM (Extended Data 1) hindered visualization of endothelial cells from the inside. Therefore, we tried it with isolated endothelial cells. This was a qualitative study as a proof-of-principle to establish that subcutaneous adipose endothelial cells can establish fenestrae *in vitro*. Unlike liver sinusoidal endothelial cells, where all endothelial cells exhibit fenestration, subcutaneous adipose tissue maybe has approx. 15% endothelial cells as fenestrated. Given that fenestrated ECs are present at such a low frequency, which might be

even further reduced *in vitro*, as fenestrae are lost over a period of time in culture, in this round, we analyzed 11 cells from 3 mice and 1 cell exhibited 174 fenestrae at the periphery.

While in visceral adipose tissue, we could detect 64 fenestrae in 1 cell out of 12 cells analyzed from 3 mice. In few images, the fenestrae-like structures are dispersed individually or hard to see, making counting somewhat arbitrary in these cases. Additionally, since these occurrences are quite rare, we decided against including statistical analysis unless we can significantly increase the sample size. However, given that SEM is not a high-throughput technique, this would require a lot of time and *in vitro* culture of cells exposes them to high levels of VEGF in the media which would not recapitulate *in vivo* conditions in adult mice. Therefore, we used this study just as a qualitative proof.

3. The authors point to increased vascular density in sWAT versus vWAT, as well as reduced density following HFD. For comparison, adipocyte size should be provided as this is an important factor to consider with regards to sWAT versus vWAT vascular density.

We thank the reviewer for raising this issue. We have normalized the vessel area to entire tissue area on the whole tissue section (both for sWAT and vWAT), therefore taking into account the size to the tissue itself (see images of whole tissue below). Therefore, any increase in adipocyte and tissue size is take into account.

Lean sWAT

Obese sWAT

4. In ED Fig 1c-d, the authors suggest activation of p53 based on representative images. These data need to be quantified.

We have done the quantifications and added it to Extended Data 1 (see figure below).

5. Despite having scRNA-seq data, the authors only show a reduction in VEGF receptor mRNA in bulk RNA analyzed via qPCRs (Fig. 4j-k). As there are significant shifts in EC populations (e.g. in fenestrated ECs), the bulk analyses are very difficult to interpret. Please provide the expression of these genes in HFD v CD per cluster / EC subtype.

In addition to qPCR data we have now added expression levels of *Vegf* receptors in sWAT and vWAT under CD and HFD conditions in Figure 4 and Extended Data 6 (see panels below).

6. Line 245: “only one report indicating the presence of fenestrated endothelium42”. This has also been shown by PMID: 36400935.

We have modified the sentence accordingly.

7. Fig. 1f – it is very difficult to read gene names due to the small fonts. Please update.

We have made the changes as requested by the reviewer. However, due to space constraints we now only show top 5 marker genes for each cluster (see below). Extensive list of all marker genes has also been provided in supplementary data.

Figure 1

8. Typos: line 128 – should be “Extended data 4f”; line 143 – should be “Extended data 3”; ED Fig 8k – there is a strange bar in the figure

We thank the reviewer for their comment and we have corrected these errors.

REVIEWERS' COMMENTS

Reviewer #1 (Remarks to the Author):

The authors have addressed all my comments and concerns. The manuscript has improved significantly since the original submission.

We thank the reviewer for the positive evaluation of our work.

Reviewer #2 (Remarks to the Author):

No further comments.

Thank you very much.

Reviewer #3 (Remarks to the Author):

We thank the authors for the thorough revisions made to the manuscript. In this revised format, the manuscript meets the requirements for a “research article” with respect to clarity in data analysis, data deposition, and code availability. The datasets previously noted as missing for a “resource article” have been verified as deposited in the repositories specified by the authors, and an extended analysis code has also been provided. The clarity of the manuscript has also notably improved, particularly in the way specific comparisons are described across different sections. The authors have also addressed all the points raised in the 'Additional Questions' section.

We thank the reviewer for the positive evaluation of our work.

The only remaining point to highlight concerns the biological significance of the findings. In response to the related comment, the authors acknowledge that “it could very well be that they [fenestrated endothelial cells in adipose tissue] do not have a role in systemic endocrine or metabolic homeostasis but a more local effect in interacting with stromal cells in the tissue.” The authors also note the current technical limitations that make specific functional studies of this subpopulation not possible due to the lack of suitable genetic tools. This is a fair point and reflects the current state of the field. Nevertheless, it does mean that the broader physiological role of these cells remains uncertain at this stage. That said, the identification of these cells in human tissue provides added value to the study. These aspects contribute to the manuscript's overall merit, even if some biological questions remain open for future investigation.

We agree with the reviewer and have highlighted this point in the discussion section (see below).

Endothelial fenestrations may facilitate the exchange of instructive and regulatory signals between both cell types⁴⁸. However, this hypothesis warrants further research. Nevertheless, the broader physiological role of this cell type in adipose tissue homeostasis is still uncertain and this remains a limitation of this study.